# Dopamine differentially modulates the size of projection neuron ensembles in the intact and dopamine-depleted striatum

**Marta Maltese[1,2], Jeffrey R March[1,2], Alexander G Bashaw[1,2], Nicolas X Tritsch[1,2]\***

[1]Neuroscience Institute, New York University Grossman School of Medicine, New York, United States; [2]Fresco Institute for Parkinson's and Movement Disorders, New York University Langone Health, New York, United States

**Abstract** Dopamine (DA) is a critical modulator of brain circuits that control voluntary movements, but our understanding of its influence on the activity of target neurons in vivo remains limited. Here, we use two-photon $Ca^{2+}$ imaging to monitor the activity of direct and indirect-pathway spiny projection neurons (SPNs) simultaneously in the striatum of behaving mice during acute and prolonged manipulations of DA signaling. We find that increasing and decreasing DA biases striatal activity toward the direct and indirect pathways, respectively, by changing the overall number of SPNs recruited during behavior in a manner not predicted by existing models of DA function. This modulation is drastically altered in a model of Parkinson's disease. Our results reveal a previously unappreciated population-level influence of DA on striatal output and provide novel insights into the pathophysiology of Parkinson's disease.

## Introduction

The neuromodulator dopamine (DA) is an essential component of basal ganglia circuits that control goal-directed behaviors. Considerable evidence in humans, primates, and rodents implicates DA in supporting motor learning and in executing vigorous movements via its actions in the striatum, the principal input nucleus to the basal ganglia (*Graybiel, 2005*; *Turner and Desmurget, 2010*; *Dudman and Krakauer, 2016*; *Klaus et al., 2019*). The striatum mainly consists of two large populations of inhibitory spiny projection neurons (SPNs) that belong to the direct and indirect pathways (dSPNs and iSPNs, respectively). The former directly inhibits output nuclei of the basal ganglia, including the internal globus pallidus and substantia nigra pars reticulata, while the latter promotes their activity indirectly by way of the external globus pallidus and subthalamic nucleus. Although long believed to exert opposite effects on movement, dSPNs and iSPNs are now known to be concurrently active and to work in concert to produce coherent sequences of voluntary movements (*Cui et al., 2013*; *Barbera et al., 2016*; *Klaus et al., 2017*; *Markowitz et al., 2018*; *Meng et al., 2018*; *Parker et al., 2018*; *Sheng et al., 2019*). How DA influences the activity of dSPNs and iSPNs in vivo to affect movement, however, remains poorly understood.

Because dSPNs express $G\alpha_s$-coupled $D_1$-type DA receptors and iSPNs $G\alpha_i$-coupled $D_2$-type DA receptors, DA is widely believed to differentially modulate both pathways and to promote imbalances that impact how the basal ganglia contribute to behavior (*Gerfen and Surmeier, 2011*; *Nelson and Kreitzer, 2014*; *Klaus et al., 2019*). Electrophysiological studies in brain slices have identified several ion channels and synaptic properties in SPNs susceptible to differential modulation by DA on both short and long timescales (*Tritsch and Sabatini, 2012*; *Zhai et al., 2019*). DA has for instance been shown to modify the intrinsic excitability of SPNs over the course of seconds to minutes, increasing and reducing the number of action potentials produced by dSPNs and iSPNs in response to somatic current injection, respectively (*Hernandez-Lopez et al., 2000*; *Ericsson et al.,*

**\*For correspondence:**
nicolas.tritsch@nyulangone.org

**Competing interests:** The authors declare that no competing interests exist.

*2013*; *Planert et al., 2013*; *Lahiri and Bevan, 2020*). These effects may account for DA's ability to transiently motivate behavior and invigorate motor actions (*Panigrahi et al., 2015*; *Hamid et al., 2016*; *da Silva et al., 2018*). DA receptor signaling also promotes the long-term potentiation and depression of cortico-striatal glutamatergic synapses in dSPNs and iSPNs (*Calabresi et al., 2007*; *Pawlak and Kerr, 2008*; *Shen et al., 2008*). Such plasticity may underlie motor learning and the formation of motor habits by striatal circuits (*Koralek et al., 2012*; *Yttri and Dudman, 2016*; *Iino et al., 2020*).

Although ex vivo studies provide compelling evidence that DA directly promotes the activation of dSPNs and impedes that of iSPNs, there are good reasons to believe that DA's modulatory effects in vivo may be more complex. First, DA does not act exclusively on SPNs. DA receptors are also found on excitatory afferents, inhibitory SPN collaterals and striatal interneurons, all of which participate in shaping striatal output (*Tritsch and Sabatini, 2012*; *Zhai et al., 2019*). Second, SPNs are constantly exposed to DA, as midbrain DA neurons fire action potentials tonically at 3–10 Hz (*Grace and Bunney, 1984*; *Cohen et al., 2012*). Given that a single pulse of DA alters the excitability of SPNs for several minutes (*Lahiri and Bevan, 2020*), additional modulation by rising DA may be occluded in vivo. Alternatively, the relatively low affinity of $D_1$ receptors for DA compared to $D_2$ receptors raises the possibility that dSPNs may only be sensitive to rising DA levels, while iSPNs may selectively respond to drops in extracellular DA (*Beaulieu and Gainetdinov, 2011*; *Iino et al., 2020*; *Lee et al., 2021*). Third, although in vivo studies comparing SPN discharge before and after DA neuron lesions in animal models of Parkinson's disease support the view that DA differentially modulates firing rates in dSPNs and iSPNs (*Mallet et al., 2006*; *Ryan et al., 2018*), they do not directly speak to DA's actions under physiological conditions, as chronic loss of DA evokes widespread homeostatic adaptations in striatal circuits (*Zhai et al., 2019*). Lastly, whether DA modulates other aspects of striatal output in addition to discharge rates, such as bursting or population-level striatal activity patterns, has been explored less extensively.

Recent imaging studies show that distinct motor actions are represented in striatum by separate groups – or ensembles – of SPNs, and that individual renditions of a given action recruit a sparse subset of SPNs from these larger action-specific ensembles (*Barbera et al., 2016*; *Klaus et al., 2017*; *Markowitz et al., 2018*). It is presently not clear how plastic these neural representations are, and if DA plays a role in sculpting them. Here, we test this possibility using two-photon $Ca^{2+}$ imaging to monitor dSPN and iSPN ensembles simultaneously in the striatum of behaving mice. We report that acute and chronic DA manipulations strongly modify the overall size of movement-related ensembles in both dSPNs and iSPNs, resulting in pathway imbalances. Importantly, DA receptor signaling does not modulate dSPN and iSPN ensembles in opposite ways. Although the size of iSPN ensembles is inversely related to DA signaling, dSPN ensembles follow an inverted U-shaped response. Chronic loss of midbrain DA neurons in a model of advanced Parkinson's disease is associated with a selective decrease in the number of active dSPNs and with dramatic changes in how movement-related dSPN ensembles respond to DA. Our results therefore reveal that, in addition to its effects on firing rates, DA regulates striatal output by reconfiguring its movement-related ensemble code.

## Results

### Simultaneous imaging of dSPN and iSPN activity

To resolve the activity of individual striatal neurons in vivo, we expressed the $Ca^{2+}$ indicator GCaMP6f (*Chen et al., 2013*) virally in the dorsolateral striatum under control of the synapsin promoter to label all neurons (*Figure 1A*). Most experiments were conducted using $Drd1a^{tdTomato}$ transgenic mice (N = 18), in which dSPNs are selectively labeled red (*Ade et al., 2011*). We subsequently performed two-photon microscopy through an imaging window chronically implanted in cortex over dorsal striatum (*Howe and Dombeck, 2016*; *Bloem et al., 2017*) in head-fixed mice locomoting on a freely rotating circular treadmill (*Figure 1B*; *Figure 1—figure supplement 1A,B*; *Video 1*). This approach allows $Ca^{2+}$ signals arising in dSPNs to be clearly distinguished from those occurring in tdTomato-negative neurons (*Figure 1C,D*), which are overwhelmingly iSPNs, and which together with dSPNs account for approximately 95% of all striatal neurons in mice (*Gerfen and Surmeier, 2011*).

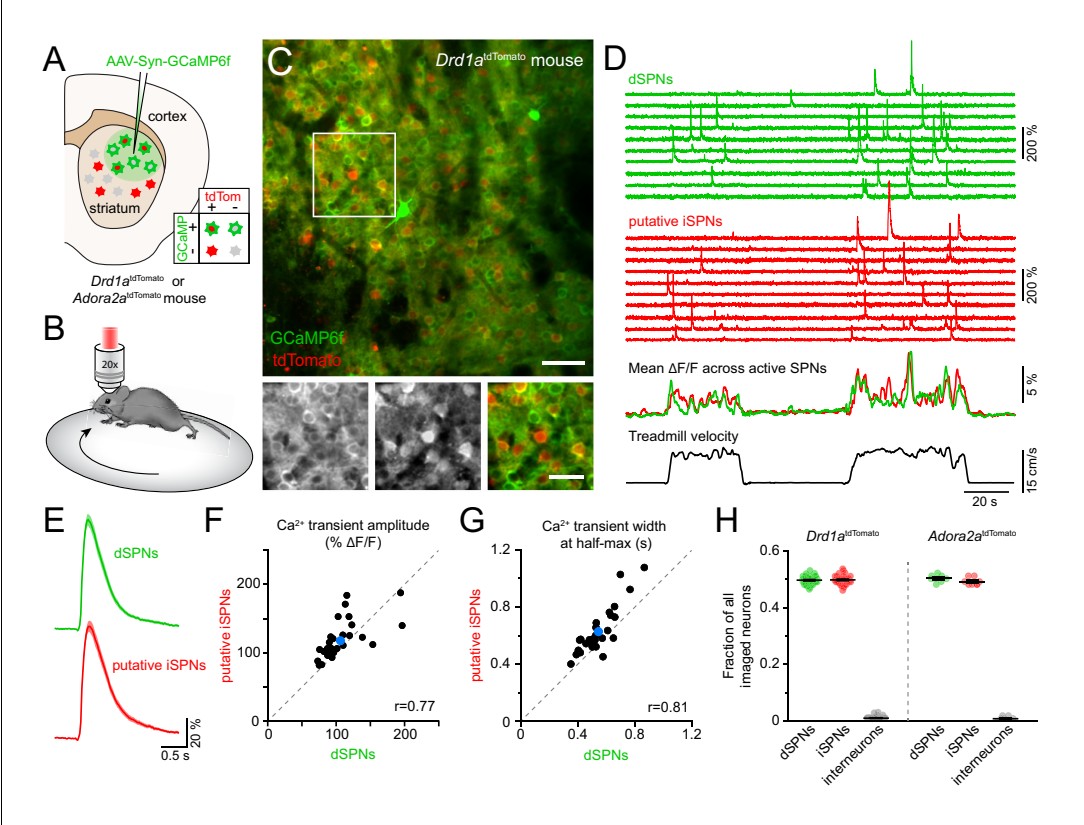

**Figure 1.** Simultaneous Ca$^{2+}$ imaging from dSPNs and iSPNs. (**A,B**) Experimental setup. GCaMP6f was virally expressed in dorsolateral striatum neurons (**A**) in mice expressing tdTomato in dSPNs (*Drd1a$^{tdTomato}$*) or iSPNs (*Adora2a$^{tdTomato}$*), and imaged by two-photon microscopy through an implanted imaging window while mice locomote on a circular treadmill (**B**). (**C**) *Top*: representative two-photon maximum projection image of dorsolateral striatum in a *Drd1a$^{tdTomato}$* mouse. Red: dSPNs, green: striatal neurons virally transduced to express GCaMP6f (scale bar: 50 µm). *Inset* shown at *bottom* in green (*left*) and red (*middle*) channels only, and composite (*right*; scale bar: 30 µm). (**D**) *Top*: example Ca$^{2+}$ fluorescence traces from dSPNs (green) and putative iSPNs (red) in a *Drd1a$^{tdTomato}$* mouse. *Middle*: Mean ΔF/F across all active dSPNs and iSPNs. *Bottom*: treadmill velocity highlighting two self-initiated locomotor bouts. (**E**) Mean Ca$^{2+}$ transient waveform (± s.e.m) imaged from active dSPNs and putative iSPNs in *Drd1a$^{tdTomato}$* mice. (**F**) Comparison of Ca$^{2+}$ transient amplitude in dSPNs versus putative iSPNs in *Drd1a$^{tdTomato}$* mice (n = 30 FOVs from 18 mice). Mean ± s.e.m is depicted in blue, and unity as dashed gray line. *r*: Spearman correlation coefficient. (**G**) Same as (**F** for Ca$^{2+}$ transient width at half-max in dSPNs versus iSPNs). (**H**) Fraction of all imaged neurons assigned to dSPNs, iSPNs and interneurons in *Drd1a$^{tdTomato}$* (n = 30 FOVs from 18 mice) and *Adora2a$^{tdTomato}$* (n = 8 FOVs in four mice) mice.

The online version of this article includes the following figure supplement(s) for figure 1:

**Figure supplement 1.** Imaging striatal activity using two-photon microscopy or photometry.

**Figure supplement 2.** Distinguishing Ca$^{2+}$ signals in dSPNs, iSPNs, and interneurons in vivo.

Self-paced forward locomotion was associated with prominent increases in fluorescence in groups of neurons both positive and negative for tdTomato (*Figure 1D*). Population Ca$^{2+}$ signals in tdTomato-positive dSPNs were similar to data obtained using photometry in mice expressing GCaMP6f selectively in dSPNs (*Figure 1—figure supplement 1C,D*). Amongst tdTomato-negative neurons, the vast majority resembled dSPNs in morphology, showed Ca$^{2+}$ transients comparable to those observed in dSPNs (*Figure 1E–G*) and displayed population activity consistent with photometric signals obtained specifically from iSPNs (*Figure 1—figure supplement 1E*), suggesting they represent iSPNs. The remaining tdTomato-negative neurons exhibited Ca$^{2+}$ signals distinct from those observed in dSPNs, including elevated baseline fluorescence (*Figure 1—figure supplement 2A–B*). These cells likely represent interneurons that sparsely populate the striatum, many of which are tonically active.

To validate our classification of tdTomato-negative cells into putative iSPNs and interneurons, we expressed GCaMP6f in neurons in a cohort of four mice in which iSPNs are selectively labeled with

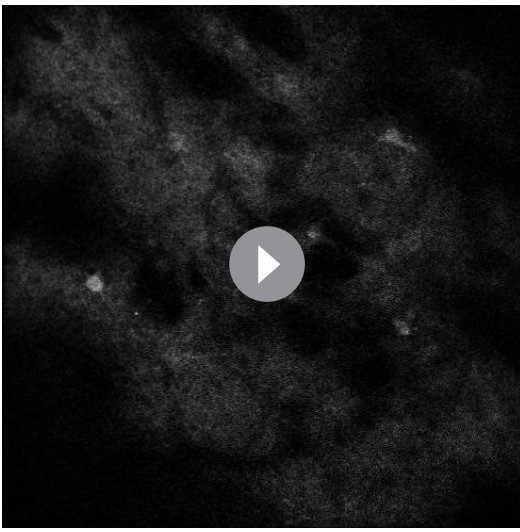

**Video 1.** Two-photon time lapse series (30 Hz frame rate) of Gcamp6f fluorescence in dorsolateral striatum as mice engage in bouts of self-paced forward locomotion. Replay speed: 1x.
https://elifesciences.org/articles/68041#video1

tdTomato either genetically (*Adora2a$^{Cre}$* transgenic mice bred to the tdTomato reporter Ai14, N = 2) or virally (*Adora2a$^{Cre}$* mice injected in dorsolateral striatum with an adeno-associated virus encoding Cre-dependent tdTomato, N = 2). We refer to this cohort as *Adora2a$^{tdTomato}$* mice henceforth. Ca$^{2+}$ signals obtained from identified iSPNs were comparable to those imaged in putative iSPNs in *Drd1a$^{tdTomato}$* mice (*Figure 1—figure supplement 2C,D*). Moreover, the small fraction of cells with elevated baseline fluorescence and distinctive activity patterns were also tdTomato-negative in *Adora2a$^{tdTomato}$* mice, indicating they are not iSPNs but likely represent interneurons. To confirm this assertion, we virally expressed Cre-dependent GCaMP6f in the striatum of *ChAT$^{Cre}$* mice (N = 2) to directly image Ca$^{2+}$ signals in cholinergic interneurons (*Figure 1—figure supplement 2E–G*). Consistent with prior reports (*Howe et al., 2019*), cholinergic interneurons exhibited elevated baseline fluorescence and movement-related Ca$^{2+}$ signals distinct from SPNs and similar to those observed in some of the tdTomato-negative cells labeled as putative interneurons in *Drd1a$^{tdTomato}$* and

*Adora2a$^{tdTomato}$* mice. Importantly, the proportions of imaged neurons assigned to dSPNs, iSPNs and interneurons in *Drd1a$^{tdTomato}$* and *Adora2a$^{tdTomato}$* mice were comparable and in agreement with anatomical estimates (*Figure 1H*), further validating our classification criteria. Our imaging approach therefore offers the ability to simultaneously monitor and compare the activity of hundreds of striatal neurons (mean ± SEM: 327 ± 13 per field of view (FOV), range: 131–442) belonging to both direct and indirect pathways with high spatial resolution during a simple behavior.

## Activity in dSPNs and iSPNs is balanced and concurrent during forward locomotion

We simultaneously imaged dSPNs and iSPNs that displayed Ca$^{2+}$ transients clearly distinguished from baseline and neuropil fluorescence during spontaneously-initiated bouts of forward locomotion (treadmill speed > 0.4 cm/s; see methods) in *Drd1a$^{tdTomato}$* and *Adora2a$^{tdTomato}$* mice. Under these conditions, intracellular Ca$^{2+}$ transients are unlikely to reflect individual spikes, but rather bursts of action potentials (*Kerr and Plenz, 2002*; *Owen et al., 2018*; *Parker et al., 2018*), which are prevalent during movement execution (*Berke et al., 2004*; *Yin et al., 2009*; *Fobbs et al., 2020*). Consistent with this, the frequency of Ca$^{2+}$ transients in dSPNs and iSPNs was exceedingly low during periods of immobility (treadmill speed < 0.2 cm/s) and rose sharply at movement onset without notable temporal offset between pathways (*Figure 2A–C*). The frequency of Ca$^{2+}$ transients increased similarly in dSPNs and iSPNs in proportion to treadmill velocity, remained elevated throughout locomotor bouts and returned to baseline shortly after movement offset.

SPNs showing Ca$^{2+}$ elevations during movement formed, over the course of several locomotor bouts, discrete groups, which we termed active SPN ensembles (*Figure 2D*). We quantified the size of active SPN ensembles as the fraction of all imaged dSPNs or iSPNs within a FOV that showed Ca$^{2+}$ transients during forward locomotion over the course of an imaging session. Active dSPN and iSPN ensembles were comparable in size, accounting for (mean ± s.e.m.) 16.9 ± 1.8% and 18.6 ± 1.8% of all imaged dSPNs and iSPNs, respectively (*Figure 2E*). For each 2 s of locomotion, approximately 10% of the overall active ensemble displayed Ca$^{2+}$ transients. Individual dSPNs and iSPNs were not systematically active across different locomotor bouts, at particular phases within movement bouts or at specific treadmill velocities (*Figure 1D* and *Figure 1—figure supplement 2B,D*), consistent with a sparse sensorimotor code in dorsolateral striatum (*Barbera et al., 2016*; *Klaus et al., 2017*; *Markowitz et al., 2018*). Importantly, the overall size of movement-related

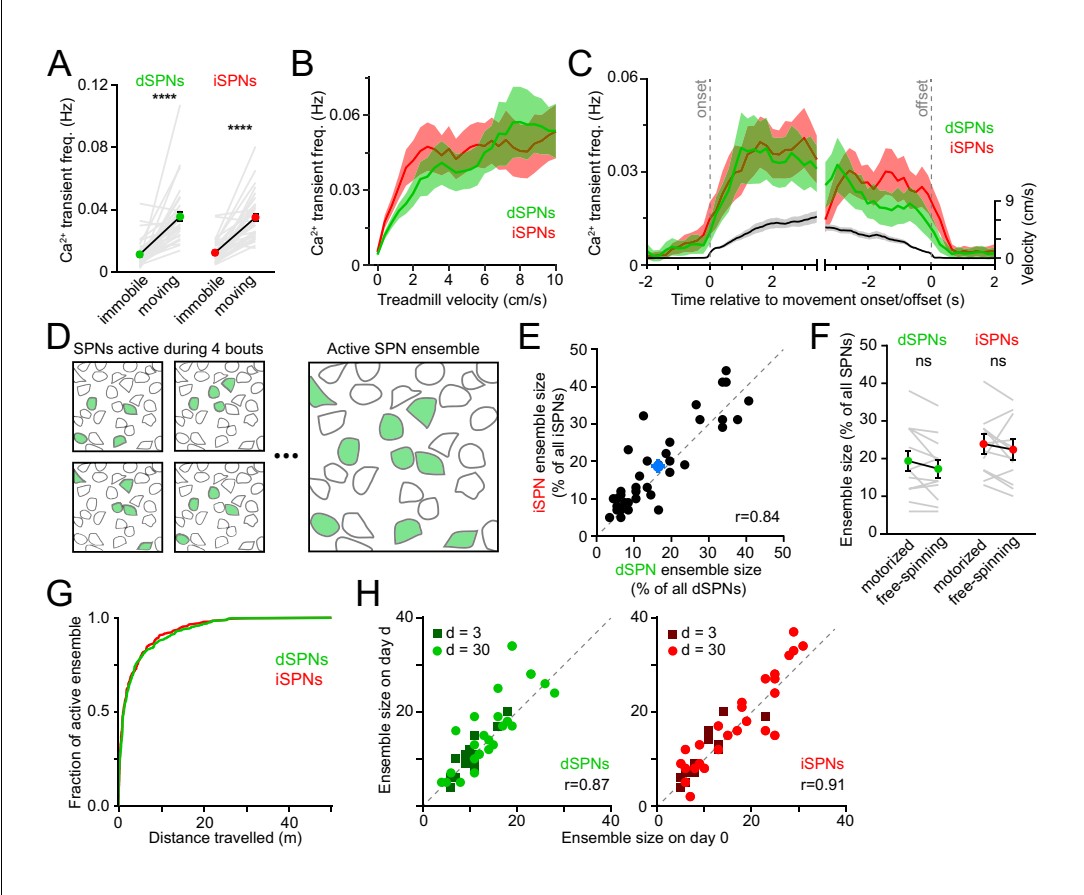

**Figure 2.** Activity in dSPNs and iSPNs is balanced during forward locomotion. (**A**) Mean Ca$^{2+}$ transient frequency per active dSPNs and iSPNs (n = 38 FOVs in 22 mice) while immobile (light gray) or spontaneously locomoting on treadmill (dark gray; p=5.5×10$^{-10}$ in dSPNs, 4.1 × 10$^{-10}$ in iSPNs, Wilcoxon signed-rank). (**B**) Mean frequency of Ca$^{2+}$ transients per active dSPN (green) and iSPN (red) at different treadmill velocities. Shaded area: s. e.m. (**C**) Same as (**B**) aligned to locomotion onset and offset, overlaid with treadmill velocity (black). (**D**) Active SPN ensembles consist of all SPNs showing Ca$^{2+}$ transients during forward locomotor bouts. Individual bouts recruit only a subset of SPNs from the overall active ensemble. (**E**) Size of active dSPN vs. iSPN ensembles, measured as the percentage of all imaged dSPNs or iSPNs exhibiting Ca$^{2+}$ transients per FOV (n = 38 FOVs from 22 mice). Mean ± s.e.m is indicated in blue and unity line in dashed gray. *r*: Spearman correlation coefficient. (**F**) Active dSPN (green) and iSPN (red) ensemble size on motorized vs. free-spinning (spontaneously-initiated locomotion) treadmills (n = 12 FOVs; dSPNs: p=0.23, iSPNs: p=0.35, Wilcoxon signed-rank). (**G**) Cumulative distribution of active dSPNs (green) and iSPNs (red) recruited as a function of distance travelled over the course of imaging sessions. (**H**) Comparison of the size of dSPN (*left*) and iSPN (*right*) ensembles imaged on separate sessions 3 days (dark dots) or a month (light dots) apart. *r*: Spearman correlation coefficient.

ensembles was not limited by distance travelled or the duration of imaging sessions, as their extent was largely captured within the first few locomotor bouts and similar values were obtained using motorized treadmills that imposed a fixed number of running bouts (*Figure 2F,G*). Ensembles were also comparable in size over time, whether imaged a few days or a month apart (*Figure 2H*). Thus, the overall activity of direct and indirect pathway SPNs – as measured by the frequency of Ca$^{2+}$ transients exhibited by individual neurons and the total number of active cells detected over an imaging session (i.e. active SPN ensembles) – is balanced, with few notable differences distinguishing dSPNs from iSPNs during self-paced forward locomotion.

## Reducing DA receptor signaling modulates the size of active SPN ensembles

DA receptor signaling modulates the firing rate of SPNs. Whether it also modulates the size of SPN ensembles representing motor actions is not known. We can conceive of two scenarios: if SPN ensembles are primarily defined anatomically by excitatory afferent connectivity, DA may alter SPN firing rates with minimal changes in ensemble size (*Figure 3A*). If, on the other hand, the extent of

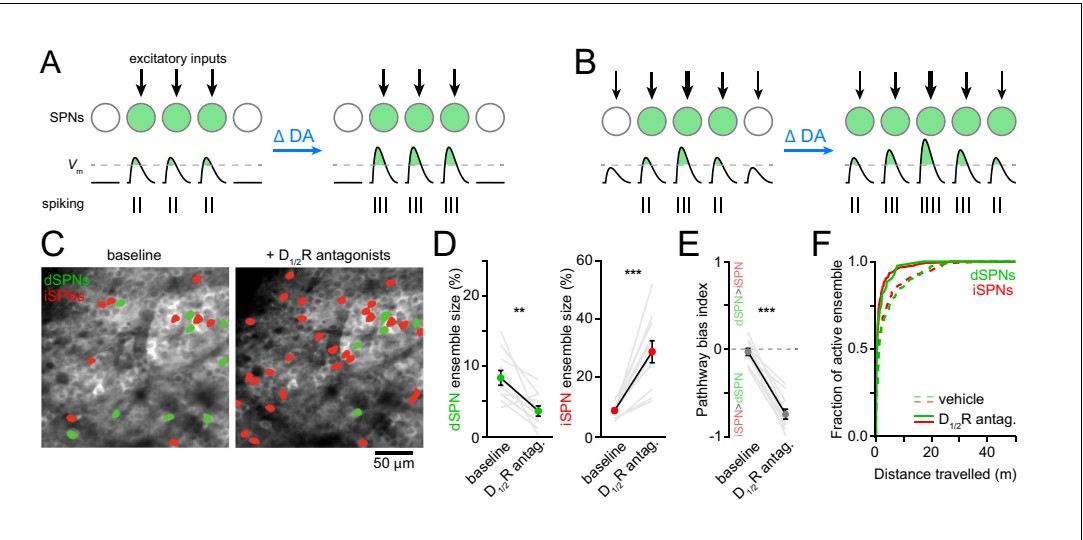

**Figure 3.** Reducing DA receptor signaling expands iSPN ensembles and shrinks dSPN ensembles. (**A,B**) DA may affect SPN firing rates without (**A**) or with (**B**) accompanying changes in active ensemble size. Active SPNs are depicted as filled circles. Movement-related excitatory inputs of varying strength (arrows) evoke action potentials (vertical bars) when postsynaptic membrane depolarizations exceed spike threshold (dashed line). In this example, changes in extracellular DA increase the strength of excitatory afferents and/or the intrinsic excitability of SPNs. (**C**) Example maximum projection image of GCaMP6f signal in dorsolateral striatum immediately before (left) and 20 min after (right) systemic administration of a cocktail of $D_{1/2}R$ antagonists (SCH23390, 0.2 mg/kg + raclopride, 1 mg/kg). dSPNs and iSPNs displaying Ca2 + transients during forward locomotion during each imaging session are highlighted in green and red, respectively. (**D**) Percentage of all imaged dSPNs (*left*) and iSPNs (*right*) showing $Ca^{2+}$ transients during locomotion before and after systemic administration of $D_{1/2}R$ antagonists (n = 11 FOVs in five mice; dSPNs: p=0.003; iSPNs: p=0.001 vs. baseline, Wilcoxon signed-rank). Mean ± s.e.m are overlaid. (**E**) Bias in the size of active dSPN and iSPN ensembles before and after $D_{1/2}R$ antagonist treatment (p=0.001 vs. baseline, Wilcoxon signed-rank). (**F**) Cumulative distribution of active dSPNs (*left*) and iSPNs (*right*) recruited as a function of distance travelled in vehicle- (dashed) and $D_{1/2}R$ antagonist-treated mice (solid), indicating that imaging sessions are sufficiently long to adequately estimate the size of active SPN ensembles.

The online version of this article includes the following figure supplement(s) for figure 3:

**Figure supplement 1.** Behavior and SPN $Ca^{2+}$ transient properties with DA receptor antagonists.

**Figure supplement 2.** Vehicle treatment does not affect striatal activity.

SPN ensembles is determined functionally by the potency of excitatory afferents and/or the excitability of SPNs, DA may exert a strong influence on ensemble size in addition to firing rates (*Figure 3B*). To distinguish between these possibilities, we first compared $Ca^{2+}$ signals in dSPNs and iSPNs before and after systemic administrations of a cocktail of $D_1$ and $D_2$-type dopamine receptor ($D_{1/2}R$) antagonists (0.2 mg/kg SCH23390 and 1 mg/kg raclopride). Although this manipulation did not impact the velocity or duration of locomotor bouts on the treadmill, it curtailed the total number of bouts that mice spontaneously initiate compared to vehicle-treated animals (*Figure 3—figure supplement 1A–C*). We therefore limited our analyses to imaging sessions on motorized treadmills to encourage locomotion and to maintain locomotor speed and distance traveled constant.

The total number of SPNs recruited by forward locomotion was significantly altered in mice treated with $D_{1/2}R$ antagonists compared to vehicle-treated controls (*Figure 3C,D* and *Figure 3—figure supplement 2A–I*). Blocking DA receptors shrunk active dSPN ensembles by half (p=0.006, Mann-Whitney; vehicle: n = 10 FOVs in five mice; DA antagonist: n = 11 FOVs in five mice), and expanded iSPN ensembles more than three-fold (p=5.7×10$^{-6}$, Mann-Whitney), resulting in a strong imbalance in favor of the indirect pathway (p=5.7×10$^{-6}$, Mann-Whitney; *Figure 3E*). This apparent dichotomous modulation is not secondary to changes in the likelihood that $Ca^{2+}$ transients are detected in dSPNs and iSPNs. Indeed, the mean amplitude of $Ca^{2+}$ transients in dSPNs and iSPNs remained unchanged (*Figure 3—figure supplement 1G–I*), and the frequency of $Ca^{2+}$ events recorded increased in both pathways, although it only reached statistical significance in iSPNs (p=0.007; Mann-Whitney vs. vehicle; *Figure 3—figure supplement 1D–F*). Moreover, the extent of active ensembles was fully captured well before the end of imaging sessions (*Figure 3F*). These data therefore indicate that reducing DA signaling evokes substantial reconfiguration of the overall

number of dSPNs and iSPNs associated with forward locomotion within minutes, and suggests that, under baseline conditions, DA continuously promotes the activation of dSPNs and represses iSPNs to maintain balance between pathways.

## Elevating DA reconfigures active SPN ensembles in a concentration-dependent manner

Given the marked effects of $D_{1/2}R$ antagonists on SPN ensemble size, we next investigated whether increasing DA signaling would exert the opposite effect. We hypothesized that dSPN ensembles would grow in size with rising DA, whereas iSPNs would be insensitive (*Iino et al., 2020*; *Lee et al., 2021*). To test this, we imaged striatal activity upon concomitant pharmacological stimulation of $D_1$ and $D_2$ receptors (n = 20 FOVs in 14 mice) while mice locomoted on motorized treadmills to mitigate effects on spontaneous behavior (*Figure 4—figure supplement 1A–C*). We examined the response of dSPNs and iSPNs to increasing concentrations of $D_{1/2}R$ agonists (SKF81297 and quinpirole, both at 0.3, 1, 3, or 6 mg/kg) in comparison to vehicle treatment.

Stimulating DA receptors markedly altered the size of SPN ensembles associated with forward locomotion, yielding a dose-dependent imbalance in favor of the direct pathway (p=$2.1\times10^{-5}$, Kruskal-Wallis test; *Figure 4B*). However, the effects on dSPN and iSPN ensembles underlying this bias varied with DA receptor agonist dose. At low concentrations (0.3–1 mg/kg), iSPN ensembles remained unchanged, whereas dSPN ensembles were either unchanged or larger, in agreement with our working hypothesis. At higher concentrations (3–6 mg/kg), $D_{1/2}R$ agonists promoted a striking decline in the prevalence of both active dSPNs and iSPNs, with iSPNs showing the strongest decrease. The drop in the number of active iSPNs suggest that iSPNs are not maximally repressed by baseline DA tone in vivo (*Marcott et al., 2014*). The reduction in the size of active dSPN ensembles was not expected. It did not stem from insufficient sampling of SPN activity during our imaging session (*Figure 4C*), or from differences in our ability to detect $Ca^{2+}$ events, as DA does not affect somatodendritic $Ca^{2+}$ transients in dSPNs (*Day et al., 2008*), and the amplitude and frequency of $Ca^{2+}$ transients were not reduced (*Figure 4—figure supplement 1D,E*).

Although DA receptors are strongly expressed in striatum, we cannot exclude the possibility that systemic administration of DA receptor agonists impact striatal activity indirectly, especially at high doses. We therefore investigated how movement-related SPN ensembles are affected by manipulations that preferentially elevate endogenous DA levels in striatum. We first considered the effects of natural rewards, which are well established to elevate extracellular DA throughout striatum (*Figure 4D*), and which were recently shown to elevate protein kinase A activity in dSPNs for 10 s of seconds (*Lee et al., 2021*). To do so, we imaged striatal activity in a cohort of water-restricted mice locomoting on a motorized treadmill before and after delivering water rewards. Water delivery selectively increased in the number of active dSPNs (*Figure 4E*) without altering $Ca^{2+}$ transient properties (*Figure 4—figure supplement 2A,B*), similar to results obtained with low concentrations of $D_{1/2}R$ agonists (*Figure 4A*).

We also considered the effects of membrane DA transporter (DAT) inhibition. Presynaptic reuptake of DA into axonal terminals is the primary mechanism through which DA neurons limit the amplitude and duration of DA transients in the striatum, and inhibiting DAT leads to strong and prolonged elevations in extracellular DA in striatum (*Sulzer et al., 2016*). We therefore compared the number of active SPNs before and after systemic administration of the DAT antagonist nomifensine (10 mg/kg). We observed effects similar to those reported with higher $D_{1/2}R$ agonist concentrations (*Figure 4A*), as nomifensine significantly decreased the size of active ensembles in both dSPNs and iSPNs (*Figure 4F*; *Figure 4—figure supplement 2C,D*). These data therefore indicate that the size of SPN ensembles recruited during forward locomotion is susceptible to modulation by rising DA, but that elevated extracellular DA levels are not invariably associated with proportional increases in the number of active dSPNs. Instead, dSPNs follow an inverted U-shaped function, where too little or too much DA negatively impacts the size of their action-related ensemble.

## Chronic DA depletion selectively limits the size of active dSPN ensembles

The degeneration of substantia nigra pars compacta (SNc) DA neurons in animal models of Parkinson's disease evokes imbalances in activity rates in dSPNs and iSPNs (*Parker et al., 2018*;

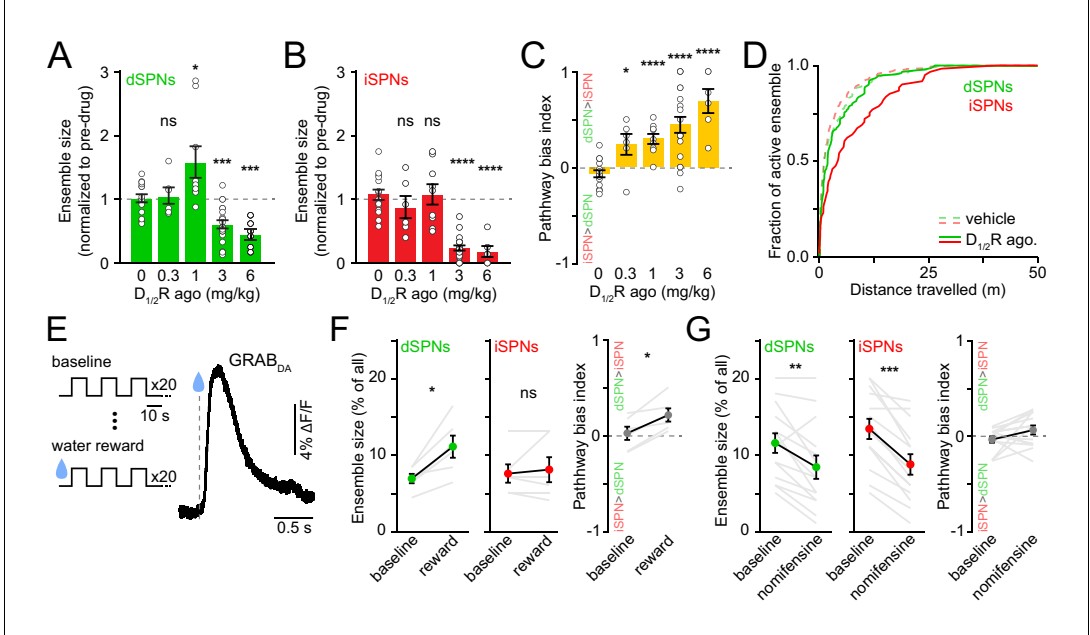

**Figure 4.** Elevating DA receptor signaling reconfigures movement-related SPN ensembles in a concentration-dependent manner. (A) Fraction of imaged dSPNs active during locomotion in mice treated with different doses of $D_{1/2}R$ agonists (SKF81297 + quinpirole) normalized to pre-drug baseline (vehicle: n = 15 FOVs; 0.3 mg/kg: p=0.92, n = 6 FOVs; 1 mg/kg: p=0.04, n = 9 FOVs; 3 mg/kg: $p=1.7\times10^{-4}$, n = 20 FOVs; 6 mg/kg: $p=2.9\times10^{-4}$, n = 6 FOVs; all vs. vehicle, Mann-Whitney). (B) Same as (A) for iSPNs (0.3 mg/kg: p=0.27, n = 6 FOVs; 1 mg/kg: p=0.92, n = 9 FOVs; 3 mg/kg: $p=3.1\times10^{-9}$, n = 20 FOVs; 6 mg/kg: $p=3.7\times10^{-5}$, n = 6 FOVs; all vs. vehicle, Mann-Whitney). (C) Bias in ensemble size between dSPNs and iSPNs (0.3 mg/kg: p=0.02; 1 mg/kg: $p=4.3\times10^{-5}$; 3 mg/kg: $p=7.8\times10^{-6}$; 6 mg/kg: $p=7.4\times10^{-5}$; all vs. vehicle, Mann-Whitney). (D) Cumulative distribution of active dSPNs (green) and iSPNs (red) recruited with travelled distance after systemic administration of vehicle (dashed) and $D_{1/2}R$ agonists (solid). (E) *Left*, experimental paradigm: mice executed blocks of three 10 s-long locomotor bouts on a motorized treadmill. The first 20 blocks were unrewarded (baseline) but the following 20 were preceded by delivery of a water reward. *Right*, example GRAB$_{DA2h}$ fluorescence (*Sun et al., 2020*) imaged from dorsolateral striatum using fiber photometry upon water reward delivery. *Inset*, mean ± s.e.m GRAB$_{DA}$ transient amplitude (n = 5 mice). (F) Active dSPN (*left*) and iSPNs (*middle*) ensemble size imaged during baseline and rewarded blocks (n = 6 FOVs in four mice; dSPN p=0.03, iSPN p>0.9, all vs. baseline, Wilcoxon signed-rank). *Right*, pathway bias index (p=0.03 vs. baseline, Wilcoxon signed-rank). Mean ± s.e.m overlaid for each group. (G) Same as (F) upon treatment with the DA transporter inhibitor nomifensine (10 mg/kg; n = 14 FOVs in six mice; dSPN p=0.005, iSPN $p=2.4\times10^{-4}$, pathway bias index p=0.68, all vs. baseline, Wilcoxon signed-rank).

The online version of this article includes the following figure supplement(s) for figure 4:

**Figure supplement 1.** Behavior and SPN Ca$^{2+}$ transient properties with DA receptor agonists.

**Figure supplement 2.** Elevating endogenous DA levels does not alter Ca$^{2+}$ transient properties.

*Ryan et al., 2018*). To examine whether chronic loss of DA also evokes changes in the size of active SPN ensembles, we injected the neurotoxin 6-hydroxydopamine (6-OHDA) into the SNc ipsilateral to the striatal hemisphere under investigation in a subset of mice (N = 12). This manipulation causes DA neurons to degenerate (*Figure 5B,C*; *Figure 5—figure supplement 1A–D*) and produces mild motor impairments as early as 24 hr – and for up to a month – after lesion (*Figure 5—figure supplement 1E–H*). Importantly, unlike the profound akinesia that results from bilateral DA denervation, hemi-lesioned mice continue to spontaneously engage in locomotor behavior, permitting the study of movement-related SPN activity within dorsolateral in the absence of DA. A separate cohort of mice (N = 6) was treated identically, except that 6-OHDA was omitted from the injection solution (sham group). These mice did not exhibit differences in the density of DA neuron cell bodies in SNc or of axons in striatum (*Figure 5C*; *Figure 5—figure supplement 1C,D*), or in striatal Ca2+ activity (*Figure 5D–F*; *Figure 5—figure supplement 2C–F*).

Lesioning DA neurons had a strong, time-dependent effect on the size of dSPN (two-way ANOVA lesion x time: $F_{2,53} = 3.66$, p=0.03; *Figure 5D*) and iSPN ensembles associated with forward locomotion ($F_{2,53} = 11.92$, $p=5.3\times10^{-5}$; *Figure 5E*), resulting in a pronounced imbalance between pathways ($F_{2,53} = 19.0$, $p=6.1\times10^{-7}$; *Figure 5F*). Within the first 24 hr, the number of dSPNs displaying Ca$^{2+}$ transients was halved, whereas active iSPNs within the same FOV more than doubled in number.

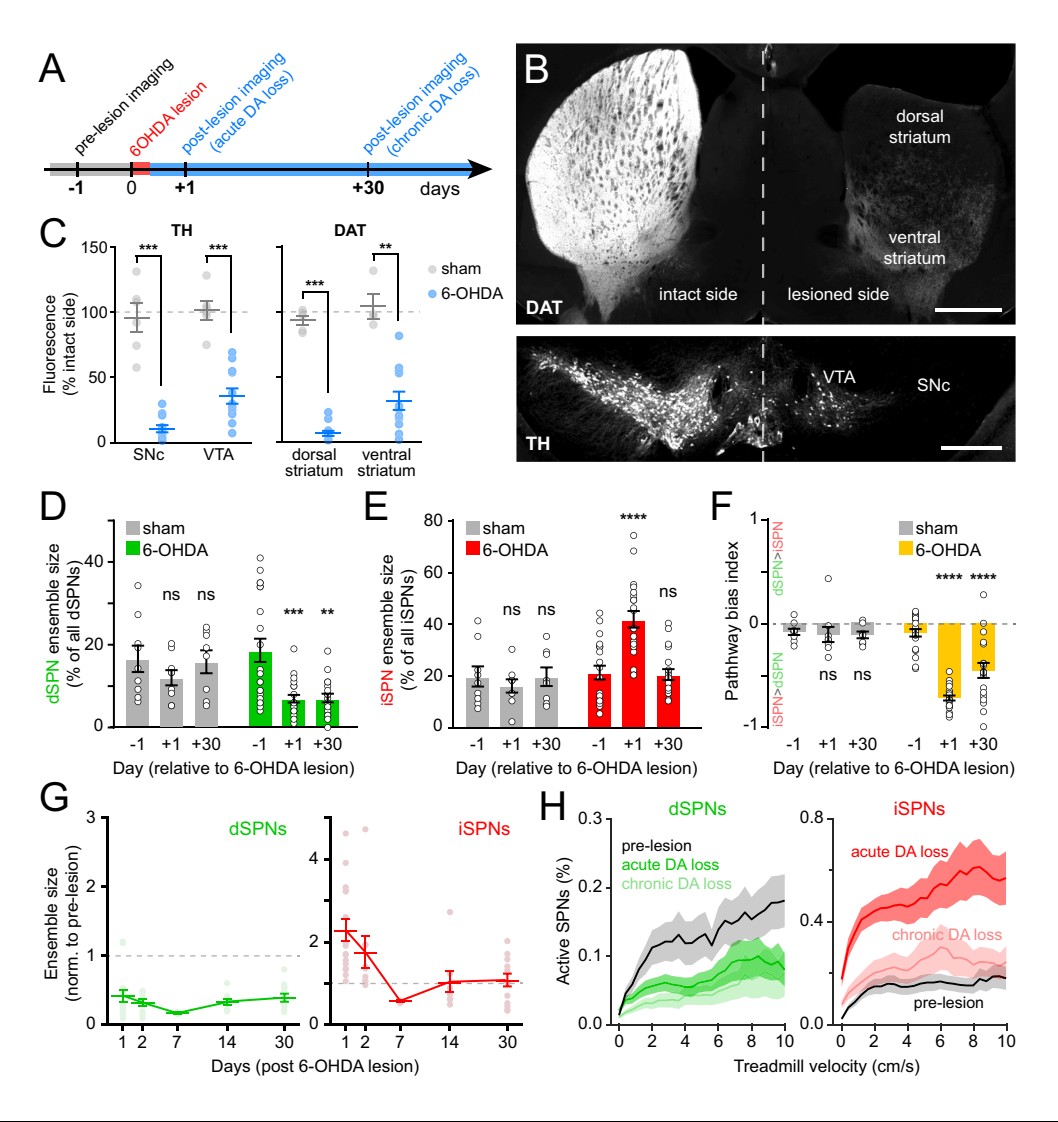

**Figure 5.** Chronic DA neuron lesions impair dSPN ensembles. (A) Experimental timeline relative to unilateral lesion of DA neurons in SNc. (B) Example coronal forebrain (*top*) and ventral midbrain (*bottom*) sections from a mouse injected with 6-OHDA into the right SNc and stained for DAT (*top*) and TH (*bottom*). VTA, ventral tegmental area. Scale bars: 1 mm (*top*), 0.5 mm (*bottom*). (C) Immunofluorescence signal for TH in SNc and VTA (*left*) and DAT in dorsal and ventral striatum (*right*) on the treated side normalized to the intact side in sham- (gray; n = 6) and 6-OHDA-lesioned mice (blue; n = 12). Mean ± s.e.m is overlaid. Asterisks depict significant difference from sham (SNc p=1.1×10$^{-4}$; VTA p=1.1×10$^{-4}$; dorsal striatum p=3.2×10$^{-4}$; ventral striatum p=0.001; Mann-Whitney). (D) Size of dSPN ensembles before (day −1), the day following (day +1) or a month after (day +30) sham (n = 9 FOVs in six mice; day 1: p=0.42; day 30: p>0.9) or 6-OHDA-mediated SNc lesions (n = 21 FOVs from 12 mice; day 1: p=1.8×10$^{-4}$; day 30: p=0.001, all vs. pre-lesion, post-hoc pairwise t-tests with Bonferroni correction). (E) Same as (D) for iSPNs (sham day 1: p=0.78; day 30: p>0.9; 6-OHDA day 1: p=8.0 × 10$^{-7}$, day 30: p>0.9, all vs. pre-lesion, post-hoc pairwise t-tests with Bonferroni correction). (F) Bias in ensemble size between dSPNs and iSPNs (sham day 1: p>0.9; day: 30 p>0.9; 6-OHDA day 1: p=3.1×10$^{-14}$, day 30: p=4.2×10$^{-5}$, all vs. pre-lesion, post-hoc pairwise t-tests with Bonferroni correction). (G) Movement-related dSPN (*left*, green) and iSPN (*right*, red) ensemble size imaged on different days after SNc lesions with 6-OHDA normalized to pre-lesion ensemble size (n = 16, 9, 2, 8, and 14 FOVs on days 1, 2, 7, 14, and 30, respectively). Note that iSPNs return to baseline values within the first 2 weeks post-lesion. (H) Percentage of imaged dSPNs (*left*) and iSPNs (*right*) displaying Ca$^{2+}$ transients in 200 ms-time bins vs. treadmill velocity during self-initiated forward locomotion before 6-OHDA lesion (black) or on day 1 (acute DA loss) and day 30 (chronic DA loss) after 6-OHDA lesion.

The online version of this article includes the following figure supplement(s) for figure 5:

*Figure 5 continued on next page*

**Figure supplement 1.** Histological and behavioral characterization of 6-OHDA-treated mice.
**Figure supplement 2.** Characterization of striatal activity in sham and 6-OHDA-treated mice.

These effects, which were similar to those observed with DA receptor antagonists (*Figure 3C–E*), were present at all recorded velocities (*Figure 5G*), as well as on a motorized treadmill (*Figure 5— figure supplement 2A*). In addition, they were not associated with significant changes in either amplitude or frequency of $Ca^{2+}$ transients in active SPNs (*Figure 5—figure supplement 2E,F*), excluding differences in behavior and signal detection as likely confounds. DA denervation therefore rapidly alters the size of dSPN and iSPN ensembles, generating a strong imbalance in favor of the indirect pathway.

To determine if homeostatic adaptations that accompany chronic DA denervation impact SPN ensembles, we also imaged striatal activity in the same cohort of mice 30 days later. Although the number of dSPNs recruited during forward locomotion remained depressed, the expansion of iSPN ensembles observed immediately after 6-OHDA treatment was no longer evident (*Figure 5D–F*). Instead, active iSPN ensembles returned to pre-lesion levels within a few days (*Figure 5G*). As with imaging sessions on day 1, we confirmed that the selective defect in dSPNs on day 30 was present across treadmill velocities (*Figure 5H*), that it was maintained on a motorized treadmill (*Figure 5— figure supplement 2A*), and that it was not associated with differences our ability to detect $Ca^{2+}$ transients or adequately capture the extent of active SPN ensembles (*Figure 5—figure supplement 2B,E,F*). Despite compensatory changes in iSPNs, the overall size of SPN ensembles recruited over the course of imaging sessions remained significantly biased toward the indirect pathway in chronically lesioned mice (*Figure 5F*). These results therefore indicate that DA depletion strongly biases striatal activity toward the indirect pathway both acutely and chronically, and point to plasticity mechanisms that normalize the number of active iSPNs, but not dSPNs, upon persistent absence of DA signaling.

## Chronic DA depletion alters how SPN ensembles respond to DA

Motor impairments in Parkinson's disease are typically managed by elevating striatal DA with L-DOPA, the metabolic precursor for DA. However, as the disease progresses and SNc degeneration becomes more severe, L-DOPA evokes abnormal involuntary movements, or dyskinesias, that severely diminish quality of life (*Cenci, 2014*; *Picconi et al., 2018*). We therefore investigated whether DA's influence on SPN ensembles changes in the striatum of mice in which DA axons are chronically lost. To do this, we first treated mice that had received unilateral 6-OHDA infusions in SNc at least one month earlier with a single dose of L-DOPA. As expected, this treatment reversed the turning bias of lesioned mice within minutes (*Figure 6—figure supplement 1A*). L-DOPA also affected SPN ensembles, albeit to a much greater extent than that observed with $D_{1/2}R$ agonists in DA-intact animals. The number of dSPNs exhibiting $Ca^{2+}$ transients during forward locomotion on motorized treadmills in DA-lesioned mice expanded by 10-fold with L-DOPA, outnumbering pre-lesion dSPN ensembles by a factor of two on average and reaching upwards of 70% of all imaged dSPNs in some fields of view (*Figure 6A,B*). In addition, the overall number of active iSPNs decreased below pre-lesion levels (*Figure 6C*), further contributing to biasing ensembles in DA-depleted mice toward the direct pathway (*Figure 6D*). These population-level effects are unlikely to stem from changes in the likelihood of detecting $Ca^{2+}$ transients in active SPNs (*Figure 6—figure supplement 1B,C*).

These results indicate that L-DOPA exerts a stronger positive modulatory influence on dSPNs in DA-depleted mice than $D_{1/2}R$ agonists do in DA-intact mice, leading to excessive recruitment of dSPNs not normally associated with forward locomotion. This discrepancy may stem from changes in how striatal circuits respond to DA following SNc neuron loss, or from differences in the pharmacological potency of these treatments. Indeed, L-DOPA's ability to elevate extracellular DA varies strongly with DA axonal density, producing disproportionately high and prolonged increases in DA in the striatum of DA-lesioned animals (*Abercrombie et al., 1990*). To distinguish between these possibilities, we directly compared the effects of L-DOPA and $D_{1/2}R$ agonists on SPN ensembles in a subset of DA-depleted mice locomoting on motorized treadmills. $D_{1/2}R$ agonists no longer provoked

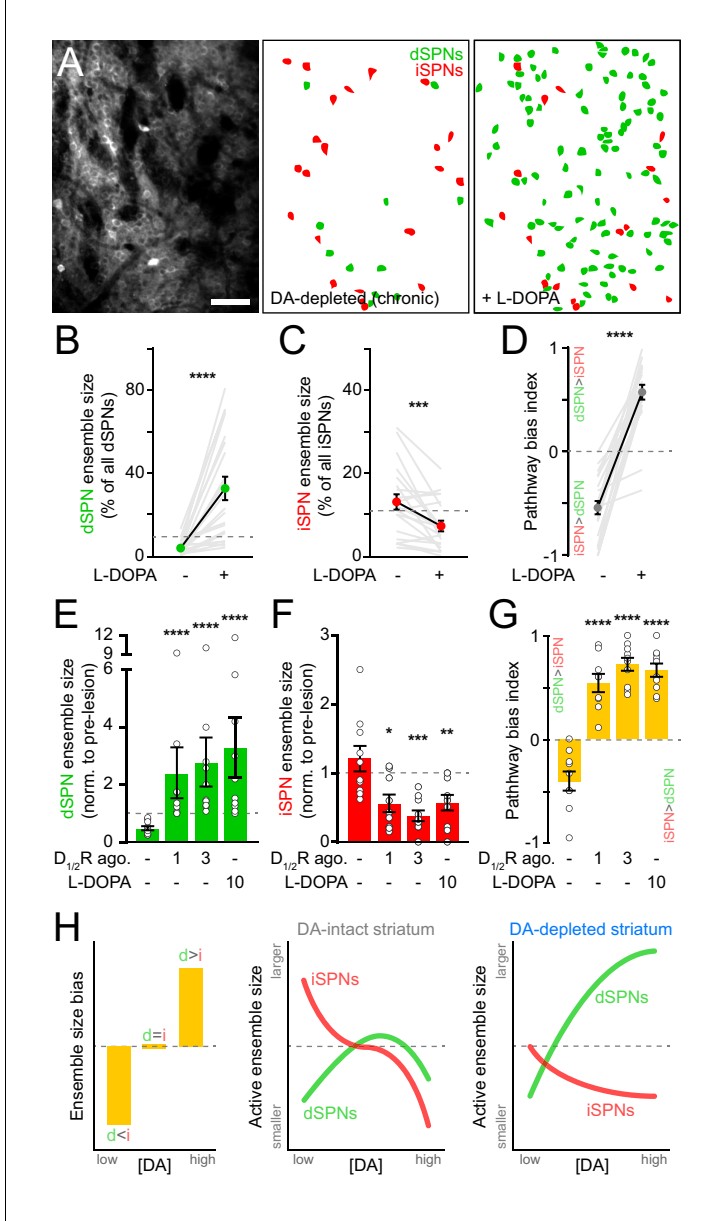

**Figure 6.** SPN ensembles respond differently to DA after chronic DA depletion. (**A**) Maximum projection image of a chronically DA-depleted dorsolateral striatum (*left*) and spatial distribution of active dSPNs (green) and iSPNs (red) ensembles imaged immediately before (*middle*) and following (*right*) systemic administration of L-DOPA. Scale bar: 50 μm. (**B**) Size of movement-related dSPN ensembles in chronically lesioned mice before and after elevating striatal DA with L-DOPA (n = 21 FOVs in 12 mice; p=1.9×10$^{-6}$, Wilcoxon signed rank). Dashed line indicates the mean size of dSPN ensembles prior to 6-OHDA lesion. (**C**) Same as (**B**) for iSPNs (n = 21 FOVs in 12 mice; p=7.0×10$^{-4}$, Wilcoxon signed rank). (**D**) Bias in the size of active dSPN and iSPN ensembles before and after L-DOPA treatment (n = 21 FOVs in 12 mice; p=9.5×10$^{-7}$, Wilcoxon signed rank). (**E**) Size of movement-related dSPN ensembles in chronically-lesioned mice (n = 10 FOVs in seven mice) imaged before (baseline) and after systemic administration of different doses of D$_{1/2}$R agonists on separate sessions (1 mg/kg: p=2.2×10$^{-5}$; 3 mg/kg: p=1.1×10$^{-5}$, all vs. baseline, Mann-Whitney) and L-DOPA (p=1.1×10$^{-5}$ vs. baseline, Mann-Whitney) normalized to the size of active dSPN ensembles prior to 6-OHDA lesion. Dashed line indicates unity. (**F**) Same as (**E**) for iSPNs (n = 10 FOVs in seven mice; 1 mg/kg: p=0.01; 3 mg/kg: p=1.3×10$^{-4}$; L-DOPA: p=8.4×10$^{-3}$; all vs. baseline, Mann-Whitney). (**G**) Bias in the size of active dSPN and iSPN ensembles before and after treatment with D$_{1/2}$R agonists or L-DOPA (n = 10 FOVs in seven mice; 1 mg/kg: p=2.2×10$^{-5}$; 3 mg/kg: p=1.1×10$^{-5}$; L-DOPA, p=1.1×10$^{-5}$; all vs. baseline, Mann-Whitney). (**H**) Diagrams summarizing observed changes in the overall balance between dSPN and

*Figure 6 continued on next page*

*Figure 6 continued*

iSPN ensembles (*left*) and the specific changes in the size of movement-related dSPN (green) and iSPN (red) ensembles at varying DA levels in the striatum of intact (*middle*) and DA-depleted mice (*right*).
The online version of this article includes the following figure supplement(s) for figure 6:

**Figure supplement 1.** Chronic DA depletion affects how striatal neurons respond to DA receptor signaling.

a decline in the number of active dSPNs in chronically lesioned mice as they did in DA-intact mice, even at high concentrations. Instead, forward locomotion recruited twice as many dSPNs as it did pre-lesion (*Figure 6E*). This expansion was comparable in magnitude to L-DOPA's, indicating that dSPNs ensembles in DA-depleted mice respond differently to DA than in intact-mice. $D_{1/2}R$ agonists also mimicked the effects of L-DOPA on iSPNs, cutting the number of active cells in half (*Figure 6F*). At all doses tested, $D_{1/2}R$ stimulation evoked a strong imbalance in favor of the direct pathway (*Figure 6G*). These data therefore indicate that DA signaling favors the recruitment of dSPNs relative to iSPNs in both DA-intact and DA-depleted mice, but that the underlying population dynamics differ profoundly (*Figure 6H*). Although dSPN ensembles are kept small with increasing DA under basal conditions, they are unrestrained in chronically lesioned mice and susceptible to dramatic expansion. At the same time, DA signaling depresses the size of iSPN ensembles, even at low levels.

## Discussion

In this study, we sought to reveal how acute and prolonged manipulations of DA signaling impact striatal output. To do so, we monitored intracellular $Ca^{2+}$ signals in dSPNs and iSPNs simultaneously using two-photon microscopy to permit the identification of subtle imbalances between direct and indirect pathways long proposed to lie at the heart of basal ganglia function and dysfunction (*Nelson and Kreitzer, 2014*; *Klaus et al., 2019*). We specifically examined changes in the size of SPN ensembles recruited during a low-dimensional locomotor behavior to facilitate comparisons across imaging sessions.

There is compelling evidence, primarily in ex vivo brain slices, that DA modulates the intrinsic excitability of SPNs and the strength of excitatory afferents to promote the activation of dSPNs and impede that of iSPNs. The net consequences of DA's actions on SPNs in vivo have almost exclusively been understood in the context of firing rates. Although this view finds experimental support (*Mallet et al., 2006*; *Ryan et al., 2018*), it does not speak to other aspects of striatal activity that DA may modulate, such as bursting or population dynamics. Our results suggest that DA exerts a strong influence on the size of movement-related SPN ensembles in ways not captured by existing models of DA function (*Figure 6H*).

Reducing DA signaling causes a strong population-level imbalance in favor of the indirect pathway by simultaneously depressing the number of dSPNs and doubling the number of iSPNs displaying movement-related $Ca^{2+}$ transients. This effect is unlikely to stem from differential modulation of voltage-gated $Ca^{2+}$ channels (*Day et al., 2008*; *Tritsch and Sabatini, 2012*) or excitatory afferents, which are largely shared between pathways (*Kress et al., 2013*; *Wall et al., 2013*; *Guo et al., 2015*). Indeed, we did not detect imbalances in either the amplitude or frequency of $Ca^{2+}$ transients in dSPNs and iSPNs. Instead, the observed ensemble changes support the view that, despite differences in the apparent affinity of $D_1$ and $D_2$ receptors for DA, basal DA levels promote the recruitment of dSPNs and limit that of iSPNs. The latter is consistent with the recent observation that protein kinase A is strongly activated in iSPNs in response to phasic dips in extracellular DA (*Lee et al., 2021*). Thus, periods of diminished DA signaling (i.e. following negative reward prediction errors) may be associated with a coincident expansion and reduction in the number of iSPNs and dSPNs responding to shared excitatory inputs with somatodendritic $Ca^{2+}$ transients, respectively.

Elevating DA signaling also affects SPN ensembles, promoting in most cases imbalance toward the direct pathway. Mild increases in DA signaling with $D_{1/2}R$ agonists or natural rewards selectively increases the size of dSPN ensembles, indicating that dSPNs remain sensitive to phasic elevations in DA. By contrast, stronger stimulation of DA receptors with higher doses of $D_{1/2}R$ agonists causes both dSPN and iSPN ensembles to shrink. A similar effect was observed when boosting endogenous

DA levels with nomifensine, which mimics the effects of stimulants like methylphenidate used in the treatment of attention deficit hyperactivity disorder. Although we cannot exclude the possibility that the observed effects reflect off-target actions of these pharmacological agents within and outside striatum, the fact that DA axons project most densely to striatum, that DA receptors are strongly expressed in striatum, and that distinct manipulations produce similar phenotypes suggest that SPN ensemble size may be directly modulated by striatal DA receptors. In addition, several electrophysiological studies have reported that $D_1R$ agonists can suppress voltage-gated $Na^+$ channels and SPN firing (*Akaike et al., 1987*; *Calabresi et al., 1987*; *Schiffmann et al., 1995*; *Hernández-López et al., 1997*; *Kravitz et al., 2010*; *Planert et al., 2013*). Collectively, these results fundamentally challenge the pervasive notion that elevating DA signaling invariably promotes the activation of dSPNs, and invite a more cautious interpretation of the effects of DA on striatal circuits. Our data show that the size of iSPN ensembles is inversely related to extracellular DA levels, whereas dSPN ensembles follow an 'inverted U' response with limited capacity for expansion with increasing DA. Interestingly, a similar relationship has been described in prefrontal cortex with DA acting on $D_1Rs$ to regulate cognitive functions like working memory (*Vijayraghavan et al., 2007*), pointing to a common mechanism of dopaminergic modulation across brain regions.

It is increasingly recognized that motor impairments in Parkinson's disease reflect aberrant activity patterns within the basal ganglia that arise not only from the loss of striatal DA, but also from subsequent homeostatic circuit adaptations (*Zhai et al., 2019*). Indeed, we observed stark differences in the activity of iSPNs after acute and prolonged depletion of DA. Whereas iSPNs initially respond to sudden DA loss with a surge in active cells similar to that evoked with $D_{1/2}R$ antagonists, movement-related iSPN ensembles returned to their pre-lesion size within a week. This observation agrees with recent work showing that persistent DA depletion in mouse models of Parkinson's disease does not evoke excessive disinhibition of the indirect pathway (*Ketzef et al., 2017*; *Parker et al., 2018*; *Ryan et al., 2018*). These results therefore indicate that iSPNs possess a remarkable capacity for homeostatic plasticity, which may stem from dendritic atrophy, spine loss, elevated inhibition from surrounding interneurons and diminished intrinsic excitability upon DA depletion (*Gittis and Kreitzer, 2012*; *Zhai et al., 2019*). By contrast, dSPN ensembles failed to functionally recover from acute loss of DA, as the number of active dSPNs remained low for up to a month after 6-OHDA treatment, despite reported increases in intrinsic excitability (*Fieblinger et al., 2014*; *Ketzef et al., 2017*). This finding echoes a previous report that many dSPNs are electrically silent in lesioned animals and more difficult to evoke spikes from (*Mallet et al., 2006*), possibly because DA depletion also weakens excitatory synapses (*Fieblinger et al., 2014*; *Parker et al., 2016*; *Ketzef et al., 2017*) and potentiates GABAergic influences onto dSPNs (*Lemos et al., 2016*). Thus, the bradykinesia that defines Parkinson's disease may result not only from elevated discharge of iSPNs during periods of immobility (*Parker et al., 2018*; *Ryan et al., 2018*), but also from the diminished ability to recruit large populations of dSPNs during movement.

DA replacement therapy with L-DOPA is the main line of treatment for Parkinson's disease. Although effective at alleviating motor impairments early in disease, L-DOPA eventually evokes dyskinesias. These abnormal involuntary movements are believed to arise in part from striatal circuit maladaptation to DA denervation beyond a critical threshold (*Cenci, 2014*; *Picconi et al., 2018*). In our 6-OHDA-lesioned animals – which model the severe SNc degeneration of advanced Parkinson's disease – restoring DA signaling with L-DOPA or $D_{1/2}R$ agonists affects SPN ensembles to an extent not seen in DA-intact mice. Specifically, active dSPN ensembles no longer showed an inverted U-shaped response to increasing DA. Instead, they grew larger than pre-lesion ensembles, reaching 60–80% of all imaged dSPNs in some preparations. This suggests that, as the disease progresses and DA degeneration worsens, the way in which dSPNs respond to DA fundamentally changes, complicating attempts at restoring balance between striatal pathways using DA. It is possible that the heightened sensitivity of DA receptors that accompanies severe DA neuron loss (*Gerfen, 2003*) underlies this population-level phenotype. We predict that large expansions of dSPN ensembles associated with motor actions will gravely compromise the ability of dSPNs to represent different movements and may explain why L-DOPA eventually induces uncontrolled choreiform and ballistic movements involving multiple body parts.

It is possible that the changes in ensemble size we reveal here reflect DA's well-established influence on SPN intrinsic excitability and firing rates (*Figure 3A,B*). The high degree of divergence and convergence of excitatory inputs to striatum (*Flaherty and Graybiel, 1994*; *Reig and Silberberg,*

2014; *Mandelbaum et al., 2019*) may allow DA's effects on intrinsic excitability, excitatory afferent strength, and firing rates to translate into a striatal population code, effectively amplifying DA's impact on downstream basal ganglia nuclei. Alternatively, DA may regulate the likelihood that SPNs respond to excitatory inputs with somatodendritic $Ca^{2+}$ spikes, a key factor governing plasticity across brain areas in health and disease (*Lerner and Kreitzer, 2011*; *Zhuang et al., 2013*; *Bittner et al., 2017*; *Nanou and Catterall, 2018*). $Ca^{2+}$ elevations in SPNs are tightly correlated with bursts of action potentials (*Kerr and Plenz, 2002*; *Owen et al., 2018*) and can be modulated by local interneurons independent of firing rate (*Owen et al., 2018*). Thus, intracellular $Ca^{2+}$ signals in SPNs may not reflect firing rates as much as the propensity to emit bursts of action potentials. As such, fluorescent $Ca^{2+}$ transients in SPNs constitute a measure of striatal activity in their own right, highlighting bursting events likely to impact $Ca^{2+}$-dependent synaptic plasticity in SPNs (*Carter and Sabatini, 2004*; *Jędrzejewska-Szmek et al., 2017*). In addition to its established role in shaping plasticity at individual synapses (*Calabresi et al., 2007*; *Pawlak and Kerr, 2008*; *Shen et al., 2008*), DA may therefore also regulate the number of neurons eligible to participate in such plasticity.

Several of our manipulations did not modify the frequency of $Ca^{2+}$ transients oppositely in dSPNs and iSPNs. We interpret this to mean that the factors that drive $Ca^{2+}$ transients in SPNs (e.g. trains of excitatory inputs) are shared between pathways and not subject to strong pathway-specific modulation by DA. This finding stands in contrast to a recent imaging study reporting differential changes in the rate of $Ca^{2+}$ events in dSPNs and iSPNs upon acute and chronic DA manipulations (*Parker et al., 2018*). This inconsistency may stem from technical differences; we monitored striatal activity in head-fixed mice consistently performing one simple movement across conditions, whereas Parker and colleagues imaged freely behaving mice performing multiple undefined movement that may vary in frequency or intensity with DA manipulations. In addition, our imaging approach clearly distinguishes changes in the frequency and amplitude of $Ca^{2+}$ transients per active neuron from changes in the overall number of active cells associated with a given action. These measures can more easily be conflated with one-photon endoscopic techniques (*Helmchen and Denk, 2005*; *Ziv and Ghosh, 2015*; *Zhou et al., 2018*). Lastly, it is important to recall that all subcortical cellular-resolution imaging studies damage cortex to gain optical access. This is a particularly important caveat in dorsolateral striatum, which receives strong excitatory inputs from the overlaying somatosensory cortex. It is therefore possible that the changes we report here mainly reflect DA modulation of thalamo-striatal circuits, which are preserved in our imaging preparation.

Taken together, our results expand classic rate-based models of DA modulation by revealing an additional dimension of DA signaling at the level of neuronal populations. We find that the size of movement-related SPN ensembles is not fixed, but rather strongly modulated by DA. Elevating DA biases striatal ensembles toward the direct pathway, while decreasing DA does the opposite. Importantly, the population dynamics underlying pathway imbalances vary with DA levels and differ in DA-intact and DA-depleted mice, prompting revisions of our understanding of DA's complex modulatory actions in vivo. Short and long-term changes in the number of SPNs that fire bursts of action potentials may constitute an integral way in which striatal circuits control the selection (*Markowitz et al., 2018*), vigor (*Panigrahi et al., 2015*; *da Silva et al., 2018*; *Yttri and Dudman, 2018*), and learning of action sequences (*Graybiel, 1998*; *Yin et al., 2009*; *Koralek et al., 2012*; *Sheng et al., 2019*; *Iino et al., 2020*). Future investigations will determine the cellular mechanisms that underlie DA's ability to modulate the size of SPN ensembles, the time course of such changes in response to synaptically released DA, and the impact of varying ensemble size on the production, learning and refinement of motor actions by the basal ganglia.

## Materials and methods

**Key resources table**

| Reagent type (species) or resource | Designation | Source or reference | Identifiers | Additional information |
|---|---|---|---|---|
| Genetic reagent (*M. musculus*) | C57BL/6J | Jackson Laboratory | RRID:IMSR_JAX:000664 | |

*Continued on next page*

*Continued*

| Reagent type (species) or resource | Designation | Source or reference | Identifiers | Additional information |
|---|---|---|---|---|
| Genetic reagent (*M. musculus*) | B6.Cg-Tg(Drd1a-tdTomato)6Calak/J; Drd1a^tdTomato | Jackson Laboratory (*Ade et al., 2011*) | RRID:IMSR_JAX:016204 | |
| Genetic reagent (*M. musculus*) | Tg(Adora2a-cre) KG139Gsat; Adora2a^Cre | GENSAT (*Gong et al., 2007*) | RRID:MMRRC_036158-UCD | |
| Genetic reagent (*M. musculus*) | Tg(Drd1a-cre)EY217Gsat; Drd1a^Cre | GENSAT (*Gong et al., 2007*) | RRID:MMRRC_030778-UCD | |
| Genetic reagent (*M. musculus*) | B6;129S6-Gt(ROSA) 26Sor^tm14(CAG-tdTomato)Hze/J; Ai14 | Jackson Laboratory (*Madisen et al., 2010*) | RRID:IMSR_JAX:007908 | |
| Genetic reagent (*M. musculus*) | B6.129S-Chat^tm1(cre)Lowl/MwarJ; ChAT^Cre | Jackson Laboratory (*Rossi et al., 2011*) | RRID:IMSR_JAX:031661 | |
| Antibody | Mouse monoclonal anti-Tyrosine Hydroxylase | Immunostar | Cat#: 22941 RRID:AB_572268 | IHC (1:1000) |
| Antibody | Rat monoclonal anti-Dopamine Transporter | Millipore | Cat#: MAB369 RRID:AB_2190413 | IHC (1:1000) |
| Antibody | Goat anti-mouse IgG Alexa Fluor 647 | Thermo Fisher Scientific | Cat#: A21236 RRID:AB_2535805 | IHC (1:500) |
| Antibody | Goat anti-rat IgG Alexa Fluor 647 | Thermo Fisher Scientific | Cat#: A21247 RRID:AB_141778 | IHC (1:500) |
| Recombinant DNA reagent | pAAV-FLEX-tdTomato | Addgene | Cat#: 28306-AAV1 RRID:Addgene_28306 | |
| Recombinant DNA reagent | pAAV-Syn-GCaMP6f-WPRE-SV40 | Addgene (*Chen et al., 2013*) | Cat#: 100837-AAV1 RRID:Addgene_100837 | |
| Recombinant DNA reagent | pAAV-hSyn-DA2m | *Sun et al., 2020* | Kindly provided by Yulong Li | |
| Chemical compound, drug | Dexamethasone | Henry Schein | Cat #: 1396050 | 4 mg/kg, I.P. |
| Chemical compound, drug | Ketoprofen | Henry Schein | Cat #: 1310364 | 10 mg/kg, S.C. |
| Chemical compound, drug | Desipramine | Tocris | Cat #: 3067 | 25 mg/kg, I.P. |
| Chemical compound, drug | Pargyline | Sigma-Aldrich | Cat #: P8013 | 5 mg/kg, I.P. |
| Chemical compound, drug | 6-OHDA | Sigma-Aldrich | Cat #: H4381 | 3 µg, I.C. |
| Chemical compound, drug | SCH23390 | Fisher Scientific | Cat #: 09-251-0 | 0.2 mg/kg, I.P. |
| Chemical compound, drug | Nomifensine | Tocris | Cat #: 1992 | 10 mg/kg, I.P. |
| Chemical compound, drug | S(-)Raclopride | Sigma-Aldrich | Cat #: R121 | 1 mg/kg, I.P. |
| Chemical compound, drug | (-)Quinpirole | Tocris | Cat #: 1061 | 0.3–6 mg/kg, I.P. |
| Chemical compound, drug | SKF81297 | Tocris | Cat #: 1447 | 0.3–6 mg/kg, I.P. |
| Chemical compound, drug | L-DOPA | Tocris | Cat #: 3788 | 10 mg/kg, I.P. |
| Chemical compound, drug | Benserazide hydrochloride | Sigma-Aldrich | Cat #: B7283 | 12 mg/kg, I.P. |
| Software, algorithm | ScanImage 5 | Vidrio Technologies (*Pologruto et al., 2003*) | RRID:SCR_014307 | Used for two-photon image acquisition |
| Software, algorithm | Custom MATLAB code | https://github.com/HarveyLab/Acquisition2P_class.git (*Driscoll et al., 2017*; *Chettih, 2019*) | | Used for two-photon image processing |
| Software, algorithm | Custom MATLAB code | https://github.com/TritschLab/TLab-2P-analysis | | Used for two-photon image analyses |
| Software, algorithm | Wavesurfer | HHMI Janelia Research Campus | https://wavesurfer.janelia.org/ | Used to register behavior and imaging data |
| Software, algorithm | Fiji | *Schindelin et al., 2012* | RRID:SCR_002285 | Used for image processing |

*Continued on next page*

*Continued*

| Reagent type (species) or resource | Designation | Source or reference | Identifiers | Additional information |
|---|---|---|---|---|
| Software, algorithm | Prism 8.0 | GraphPad | RRID:SCR_002798 | Used for plotting and statistical analyses |
| Other | Mounting medium with DAPI | Fisher Scientific | Cat #: P36931 | |

## Animals

All procedures were performed in accordance with protocols approved by the NYU Langone Health Institutional Animal Care and Use Committee (protocol # 170123). Mice were housed in group before surgery and singly after surgery under a reverse 12 hr light-dark cycle (dark from 10 a.m. to 10 p.m.) with ad libitum access to food and water. *Drd1a*[tdTomato] transgenic mice (stock #: 016204) and *ChAT*[Cre] knock-in mice (stock #: 031661) were purchased from The Jackson Laboratory and bred with C57BL/6J wild type mice (stock #: 000664). Transgenic mice expressing Cre selectively in iSPNs (*Adora2a*[Cre]; KG139) or dSPNs (*Drd1a*[Cre]; EY217) (*Gong et al., 2007*) were generously provided by Chip Gerfen (NIH) maintained on a C57BL/6J background. Control experiments in *Figure 1—figure supplement 2* were performed using either offspring of *Adora2a*[Cre] mice bred to a tdTomato reporter (Ai14; The Jackson Laboratory, Stock #: 007908; *Madisen et al., 2010*) or *Adora2a*[Cre] mice injected in dorsolateral striatum with an adeno-associated virus (AAV) encoding Cre-dependent tdTomato (Addgene; #28306-AAV1). Experiments were carried out using both male and female mice heterozygous for all transgenes at 8–24 weeks of age.

## Surgery

Mice were administered dexamethasone (4 mg/kg, intraperitoneal) 1–2 hr prior to surgery. They were then anaesthetized with isoflurane, placed in a stereotaxic apparatus (Kopf Instruments) on a heating blanket (Harvard Apparatus) and administered Ketoprofen (10 mg/kg, subcutaneous). The scalp was shaved and cleaned with ethanol and iodine solutions before exposing the skull. A custom titanium headpost was implanted over lambda using C and B metabond (Parkell) to allow head fixation. To achieve widespread viral expression of GCaMP6f in striatum, 100 nl of AAV1-Syn-GCaMP6f-WPRE-SV40 was injected alone or along with AAV1-FLEX-tdTomato 1.7 mm below dura at a rate of 100 nl/min (KD Scientific) into the right dorsolateral striatum at four locations (anterior/lateral from bregma, in mm): 0.7/1.7; 0.7/2.3; 1.3/1.7, and 1.3/2.3. Injection pipettes were left in place for 5 min before removal. A 3 mm craniotomy was then drilled (centered at 1.0 mm anterior and 2.0 mm lateral from bregma) and cortical tissue was aspirated until the corpus callosum lying above the striatum was exposed, as described previously (*Howe and Dombeck, 2016*; *Bloem et al., 2017*). A custom nine gauge thin-walled stainless-steel cannula (Microgroup; 2.3 mm in height) sealed at one end with a 3 mm glass coverslip (Warner Instruments) using optical glue (Norland #71) was placed above the striatum and cemented to the skull using C and B metabond (see *Figure 1—figure supplement 1A,B*). For fiber photometry imaging, 200 nl of AAV1-Syn-FLEX-GCaMP6f-WPRE-SV40 or AAV9-hSyn-DA2m (kindly provided by Dr. Yulong Li) was injected 2.2 mm below dura into the right dorsolateral striatum (anterior/lateral from bregma: 1.0/2.0) and a 400 μm optic fiber (Thorlabs, FP400URT) housed in a ceramic ferrule (Thorlabs, CFLC440-10) was implanted 0.2 mm above the injection site using C and B metabond. Mice were allowed to recover in their cage for 2 weeks before head-fixation habituation, treadmill training and imaging.

## 6-OHDA lesions

For experiments characterizing the effects of SNc neuron loss on striatal activity, a second stereotaxic surgery was performed under isoflurane anesthesia after baseline imaging sessions. Desipramine (25 mg/kg) and pargyline (5 mg/kg) were administered intraperitoneally prior to surgery to increase the selectivity and efficacy of 6-OHDA lesions (*Thiele et al., 2012*). A small craniotomy was performed above the SNc ipsilateral to the imaged striatum (−3.1 mm posterior from bregma, 1.3 mm lateral) and 3 μg of 6-OHDA (total volume: 200 nl) was injected in SNc (4.2 mm below dura) at a rate ~ 100 nl/min. Sham-lesioned mice were treated identically except that 6-OHDA was omitted from the injected solution (0.2% ascorbic acid in 0.9% sterile NaCl solution). Mice were randomly

assigned to each group. To evaluate motor impairments, mice were placed in a plus maze consisting of four identical closed arms (length x width x wall height: 45 cm × 7 cm × 15 cm) at 90° to each other and monitored from above with a near infrared camera (Basler; acA2000-165um) for 10 min. All arms of the maze were closed (i.e. featured 15 cm-tall walls) to avoid anxiety-related behaviors. The number of ipsiversive, contraversive and straight choices upon reaching the center of the maze were automatically quantified in MATLAB (Mathworks) and verified visually. Data were expressed as a turning bias index, defined as the difference between ipsiversive and contraversive turns, divided by the total number of turns.

## Immunohistochemistry

Mice were deeply anesthetized with isoflurane and perfused transcardially with 4% paraformaldehyde in 0.1 M sodium phosphate buffer. Brains were post-fixed for 1–3 days, sectioned coronally (50–100 μm in thickness) using a vibratome (Leica; VT1000S) and processed for immunofluorescence staining for tyrosine hydroxylase (Immunostar; 22941, 1:1000) and dopamine transporter (Millipore; MAB369, 1:1000) using standard methods. Brain sections were mounted on superfrost slides and coverslipped with ProLong antifade reagent with DAPI (Molecular Probes). Endogenous tdTomato and GCaMP6 fluorescence were not immuno-enhanced. Whole sections were imaged with an Olympus VS120 slide scanning microscope and high-resolution images of regions of interest were subsequently acquired with a Zeiss LSM 800 confocal microscope. TH and DAT immunofluorescence were quantified in ImageJ (NIH) by measuring mean pixel intensity in similarly sized regions of interest in VTA, SNc, dorsal striatum and ventral striatum in both intact and lesioned hemispheres. After subtracting background signal from adjacent unstained brain regions, fluorescence intensity values in the lesioned hemisphere were expressed as a percentage of values measured on the intact side. Three measurements were obtained from each region of interest along the anterior-posterior axis and averaged together for each mouse.

## Reagents

Drugs (all from Tocris, unless specified otherwise) were reconstituted and stored according to the manufacturers' recommendations. 6-Hydroxydopamine (6-OHDA; Sigma-Aldrich) was dissolved freshly into sterile 0.9% NaCl and 0.2% ascorbic acid immediately prior to administration to minimize oxidation. Working concentration aliquots of desipramine (25 mg/kg), pargyline (Sigma-Aldrich; 5 mg/kg), S(-)raclopride (Sigma-Aldrich; 1 mg/kg), SCH23390 (Fisher Scientific; 0.2 mg/kg), nomifensine (10 mg/kg), (-)quinpirole (0.3–6 mg/kg), SKF81297 (0.3–6 mg/kg), and L-DOPA (10 mg/kg) were prepared daily in sterile physiological saline and administered intraperitoneally at minimum 10 min prior to experimentation. L-DOPA was co-administered along with the peripheral DOPA decarboxylase inhibitor benserazide hydrochloride (Sigma-Aldrich, 12 mg/kg).

## Two-photon imaging

Imaging was performed through a 20X long working-distance air objective (Edmund Optics; #58373) on a galvo-resonant scanning microscope (Thorlabs; Bergamo-II) equipped with GaAsP photomultiplier tubes and under the control of ScanImage five software (Vidrio Technologies). GCaMP6f and tdTomato were excited a using a pulsed dispersion-compensated Ti:Sapphire laser (Coherent; Chameleon Vision II) tuned to 940 nm (40–80 mW at the sample). Mice were head-fixed above a freely rotating circular treadmill consisting of a 6' plastic saucer (Ware Manufacturing) mounted on a rotary encoder (US Digital, Serial #: MA3-A10-125-B). Motorized, fixed-distance and velocity trials were conducted by connecting a integrated servo motor (Teknic Clearpath: CPM-MCVC-2310S-RQN) to the treadmill to impose 10 s-long fixed-speed (10 cm/s) running bouts every 20 s. Water-rewards (~0.1 ml) were delivered through a spout controlled by an inaudible solenoid (The Lee Company, LHQA0531220H) at 60 s intervals between motorized running bouts. Mouse posture was simultaneously monitored using a near infrared camera (Basler; acA2000-165um), with each video frame triggered by the two-photon imaging frame clock. The latter was recorded in Wavesurfer (https://www.janelia.org/open-science/wavesurfer) to synchronize imaging and behavioral data. Striatal fields of view (500 × 500 μm; 1–3 per mouse) near the center of the imaging window were selected based on the expression of GCaMP6f and tdTomato, and were continuously imaged with a resolution of 512 × 512 pixels at 30 Hz frame rate for a minimum of 20 min each. Bleaching of GCaMP6f was

negligible. The same fields of view were imaged multiple times on non-consecutive days using alignment based on blood vessels and a reference image.

## Fiber photometry

Excitation light (Thorlabs; M470F3) was passed through a fluorescence mini-cube (Doric, FMC5-E1-460/490) and 400 μm fiber optic patch cord connected to the mouse via a zirconia sleeve. Emission was collected through the same patch cord and mini-cube to a femtowatt photoreceiver (Newport, 2151). Photometry signals were digitized at 2 kHz using a National Instruments acquisition board (USB-6343), recorded with Wavesurfer, and low-pass filtered and down-sampled to 30 Hz in MATLAB (Mathworks).

## Image processing

Time series of two-photon images were concatenated and motion corrected, and $Ca^{2+}$ fluorescence traces were extracted from individual neurons with minimal neuropil contamination using a custom MATLAB (Mathworks) pipeline described in detail in *Driscoll et al., 2017* and kindly made available by Chris Harvey (Harvard Medical School; https://github.com/HarveyLab/Acquisition2P_class.git; *Chettih, 2019*). Briefly, putative cell bodies were selected manually in the mean intensity image. Highly correlated and spatially-contiguous fluorescence sources within 30 μm of the selected pixel were then identified automatically based on the correlation structure of the pixel time series to delineate the spatial footprint of active neuronal processes. Fluorescence time series were computed by averaging across all pixels within that footprint and were manually classified into cell bodies, processes (not analyzed), and background (neuropil). This approach is advantageous over manually defined regions of interest as it segregates dendritic processes that overlap with somata if they exhibit distinct fluorescence time series. Each putative cell was paired with a neighboring background source, which was subtracted from the cell body's fluorescence time series. Segmentation and neuropil subtraction were manually verified for each cell and adjusted when necessary using a graphical user interface to obtain clean neuropil-subtracted fluorescence traces without apparent negative-going transients. GCamp6f-labelled cell bodies were manually labeled as dSPNs or iSPNs based on tdTomato fluorescence in the mean intensity image in *Drd1a*[tdTomato] and *Adora2a*[tdTomato] mice, respectively, and tdTomato-negative cells were either labeled as putative iSPNs (in in *Drd1a*[tdTomato] mice), putative dSPNs (*Adora2a*[tdTomato] mice), or putative interneurons based on cell morphology, baseline $Ca^{2+}$ fluorescence intensity and $Ca^{2+}$ transient kinetics (*Figure 1* and *Figure 1—figure supplement 2*) by an experimenter blind to experimental conditions.

## Image and behavior analyses

Quantification of imaging and behavioral data was carried out in MATLAB using custom-code available online (https://github.com/TritschLab/TLab-2P-analysis; *TritschLab, 2021*; copy archived at swh:1:rev:a32d12e2dd10eb0ff7510fbbdc83aea7cf3c7356). $Ca^{2+}$ transients were detected using the built-in MATLAB findpeaks() function on each cell's neuropil-subtracted fluorescence trace smoothed with a 150 ms window if they exhibited a minimum peak height and minimum peak prominence of five standard deviations greater than baseline, and a minimum width of 140 ms at half peak prominence. These criteria are intentionally stringent so as to exclude events not clearly resolved from baseline fluorescence at the cost of underestimating $Ca^{2+}$ transient frequency per SPN. All imaged neurons displaying at minimum one $Ca^{2+}$ transient were deemed active. Individual fields of view were only selected for longitudinal imaging and quantification if at least five active dSPNs and five active iSPNs were observed at baseline in self-initiated and motorized locomotor trials, and if mice traveled at minimum five meters in self-initiated trials to ensure adequate estimation of SPN ensemble size and mean SPN $Ca^{2+}$ transient frequency. For each imaging session and field of view (FOV), we report the percentage of dSPNs and iSPNs that display $Ca^{2+}$ transients (i.e. active ensemble size), the frequency of $Ca^{2+}$ transients recorded per active dSPNs or iSPNs while moving or immobile averaged across each FOV, and the mean amplitude of $Ca^{2+}$ transients recorded in all active dSPNs or iSPNs averaged for each FOV. Active ensemble size is calculated as the percentage of all imaged dSPNs or iSPNs in a given FOV that display $Ca^{2+}$ transients over the course of an imaging session. Activity in dSPNs and iSPNs was compared for each imaging session and FOV by computing a bias index, consisting of the difference in ensemble size, mean $Ca^{2+}$ transient frequency or amplitude

between dSPNs and iSPNs, divided by their sum. Treadmill velocity was extracted from positional information provided by the rotary encoder, down-sampled to 30 Hz and aligned to the two-photon imaging frame rate. Immobility is defined as any period of time beginning at least 0.5 s after tread-mill velocity decreases below 0.2 cm/s, lasting at minimum 4 s, and ending 0.5 s before treadmill velocity exceeds 0.2 cm/s again and for which no postural movement is detected using the infrared camera. Movement bouts are defined as any period of time when absolute treadmill velocity exceeds 0.4 cm/s for a minimum of 4 s, and is preceded and followed by 4 s of immobility. Move-ment onsets and offsets are the first and last time points, respectively, at least two standard devia-tions away from treadmill velocity measured while immobile. To relate $Ca^{2+}$ transient frequency to treadmill velocity, we recorded the instantaneous treadmill velocity at each $Ca^{2+}$ transient onset and divided the total number of $Ca^{2+}$ transients per velocity (binned at 0.4 cm/s) per SPN by the amount of time spent by a mouse at that velocity over the course of an imaging session. To relate the per-centage of active SPNs to velocity, we averaged treadmill velocity in 200 ms bins, determined the fraction of all imaged dSPNs or iSPNs active within each bin and calculated the mean percentage of active dSPNs and iSPNs at different treadmill speeds (binned to 0.4 cm/s). To align $Ca^{2+}$ transient frequency to movement onset/offset, we calculated the mean number of $Ca^{2+}$ transients imaged per FOV per 200 ms-long bin around all locomotor bout onsets/offsets divided by the total number of active SPNs and bin duration.

## Statistical analyses

For each experiment, data (reported in text and figures as mean ± s.e.m) were collected at minimum from two separate cohorts of mice (biological replicates) and were compared in Prism 8 (Graphpad) using non-parametric Mann-Whitney U test for independent groups and Wilcoxon signed-rank test for matched samples, and Two-way ANOVAs (mixed-effect model) for multiple comparisons, as indi-cated in text and figure legends. Significant interactions were followed-up with within-group t-tests corrected for multiple comparisons (Bonferroni) between baseline and test data. p Values smaller than 0.05 were considered statistically significant and assigned the following nomenclature in Fig-ures: $*p<0.05$, $**p<0.01$, $***p<0.005$ and $****p<0.001$. Reported n values represent the number of fields of view imaged from N mice.

## Acknowledgements

We thank Adam Carter, Un Kang, Michael Long, Adam Mar, Margaret Rice, Dick Tsien and members of the Tritsch laboratory for comments on the manuscript. We thank the GENIE Program and the Janelia Research Campus, specifically V Jayaraman, R Kerr, D Kim, L Looger, and K Svoboda for making GCaMP6f available.

## Additional information

### Funding

| Funder | Grant reference number | Author |
| --- | --- | --- |
| National Institutes of Health | R00NS087098 | Nicolas X Tritsch |
| National Institutes of Health | DP2NS105553 | Nicolas X Tritsch |
| Alfred P. Sloan Foundation | | Nicolas X Tritsch |
| Dana Foundation | | Nicolas X Tritsch |
| Whitehall Foundation | | Nicolas X Tritsch |
| Leon Levy Foundation | | Nicolas X Tritsch |
| Marlene and Paolo Fresco In-stitute | | Marta Maltese Nicolas X Tritsch |

The funders had no role in study design, data collection and interpretation, or the decision to submit the work for publication.

## Author contributions
Marta Maltese, Conceptualization, Data curation, Formal analysis, Investigation, Writing - original draft, Writing - review and editing; Jeffrey R March, Data curation, Software, Formal analysis, Investigation, Writing - original draft; Alexander G Bashaw, Investigation; Nicolas X Tritsch, Conceptualization, Formal analysis, Supervision, Funding acquisition, Investigation, Writing - original draft, Project administration, Writing - review and editing

## Author ORCIDs
Marta Maltese (iD) https://orcid.org/0000-0002-9084-0411
Nicolas X Tritsch (iD) https://orcid.org/0000-0003-3181-7681

## Ethics
Animal experimentation: This study was performed in strict accordance with the recommendations in the Guide for the Care and Use of Laboratory Animals of the National Institutes of Health. All procedures were carried out according to protocols approved by the NYU Langone Health Institutional Animal Care and Use Committee (protocol #170123).

## Decision letter and Author response
Decision letter https://doi.org/10.7554/eLife.68041.sa1
Author response https://doi.org/10.7554/eLife.68041.sa2

# Additional files

## Supplementary files
• Transparent reporting form

## Data availability
Source data and code used for analyses are available online (https://github.com/TritschLab/Maltese-et-al-2021 [copy archived at https://archive.softwareheritage.org/swh:1:rev:69ef87387f0e588c71-ba8dc25fdb91f936c609e0] and https://github.com/TritschLab/TLab-2P-analysis [copy archived at https://archive.softwareheritage.org/swh:1:rev:a32d12e2dd10eb0ff7510fbbdc83aea7cf3c7356]).

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
