## [Decision Letter]

**Acceptance summary:**

This paper is of broad interest to neuroscientists studying Parkinson's disease, dopamine (DA) modulation and the striatum. The study provides a unique view of how dopamine modulates striatal activity in vivo by investigating effects on the size of movement-related projection neuron ensembles in DA-intact and DA-depleted mice. Whereas prior studies have focused on DA modulation of SPN excitability and firing rates, this paper introduces highly novel concepts about how dopamine modulation of the number of activated neurons (ensemble size) may influence direct/indirect pathway balance and its potential significance for Parkinson's disease.

**Decision letter after peer review:**

Thank you for submitting your article "Dopamine differentially modulates the size of projection neuron ensembles in the intact and dopamine-depleted striatum" for consideration by *eLife*. Your article has been reviewed by 3 peer reviewers, one of whom is a member of our Board of Reviewing Editors, and the evaluation has been overseen by Gary Westbrook as the Senior Editor. The following individual involved in review of your submission has agreed to reveal their identity: Ilana B Witten (Reviewer #3). The reviewers have discussed their reviews with one another, and the Reviewing Editor has drafted this to help you prepare a revised submission.

Essential revisions:

As you can see in the comments below, there was great enthusiasm for your study by all three reviewers. The reviewers do not ask for new experiments, but please address the requests listed below to address because they will enhance clarity of presentation and interpretation. The full reviews are also included below for your consideration. For Reviewer 2 #3 item concerning evaluation of dyskinesia data, it is not required, but optional to address.

1. Reviewer 1: Items 1-3

2. Reviewer 2: Items 1,2,4,5

*Reviewer #1 (Recommendations for the authors):*

The claims of the paper are largely well supported by current data. However, several aspects of data analysis could be done or clarified to further support the conclusions.

1. The dSPN U-shaped curve response is a potentially novel finding, but appears to be supported by a single data set at 1mg/kg in Figure 4A. These data also includes values that are non-normally distributed and close to outlier-like values. Was this finding replicated in a second cohort?

2. The authors focused on the analysis of ensemble size, firing rate and frequency of activated SPNs during forward locomotion. Presumably, locomotion arises from collaborative activation of certain SPN subsets and suppression of other subsets. Data in rearward locomotion and transitions should be readily available. Its evaluation would be important to bolster the findings, or reveal unexpected variances to the current interpretations.

3. Similarly, when only "total locomotion" is presented as the outcome measure, sub analyses should be performed on richer features of the movement data and presented/discussed to ensure that differential findings are not equally or better explained by differences in forward, reverse, or stalled locomotion or velocity.

*Reviewer #2 (Recommendations for the authors):*

1. While D1 receptors, at least within the striatum, are relatively selective in their expression on dSPNs, D2 antagonists or agonists may act on iSPNs themselves, or local microcircuit elements, including cholinergic interneurons, dopamine terminals, and potentially local glutamatergic and GABAergic synapses. The authors allude to the glutamatergic inputs in the discussion, but one question in my mind was how much of the results here could be related to D2 autoreceptors on dopamine terminals. For example, D2 antagonists presynaptically increase striatal dopamine release, and D2 agonists decrease dopamine release (Palij et al., 1990 and others). At the very least, it would be worth discussing the potential mechanisms of the pharmacology.

2. Related to the above comment, though some of the differences between healthy and 6-OHDA treated animals in their response to dopamine drugs may be related to chronic changes in synaptic connectivity or intrinsic properties, some may relate to the loss of one of the targets of these drugs: the dopamine terminals. Might the effects of D2 antagonists in healthy animals be an interplay between direct effects on SPNs themselves, plus changes in dopamine release mediated by actions on dopamine terminals? And one of the reasons it's so different in 6-OHDA treated animals is that now this element has been removed?

3. The authors refer to L-DOPA-induced dyskinesia in their discussion, and compare their findings to published work with L-DOPA-induced dyskinesia, but this behavioral phenomenon was not mentioned/quantified in the text. Based on my understanding of the literature, the severity of dopamine depletion they induced (Figure 5B/C) combined with a dose of 10 mg/kg of L-DOPA would likely induce dyskinesia (eg Smith et al., 2012). Did their animals show dyskinesia? And if so, how did this relate to the patterns of neural activity reported?

4. For interpretability and clarity, it would be helpful to include details of some major methods in the Results text and/or main Figure legends. For example, the authors should indicate in the Results and/or Figure legends that they used both an A2a-Cre x Ai14 cross and a viral approach to directly label iSPNs, that they used a cocktail of SCH23390 and raclopride to block D1/2 type dopamine receptors, or a cocktail of SKF81297 and quinpirole to activate D1/2 type dopamine receptors.

5. It is unclear between the Figure and methods text what concentration of L-DOPA was used – the figure itself has "10" listed underneath (Figure 6E-G), but the methods list 12 mg/kg (which may instead refer to benserazide dosing?).

*Reviewer #3 (Recommendations for the authors):*

I thought this was a nice study. There have been few studies to examine DA modulation of D1R and D2R activity and I thought the dual population imaging was cool, I was interested to see the large population level effects that mostly recapitulated expectations about the two pathways, as well as to see the inverted-U effect on D1Rs which was interesting and I was not aware of that being observed in striatum before. I didn't have any major concerns.

---

## [Author Response]

Reviewer #1 (Recommendations for the authors):The claims of the paper are largely well supported by current data. However, several aspects of data analysis could be done or clarified to further support the conclusions.1. The dSPN U-shaped curve response is a potentially novel finding, but appears to be supported by a single data set at 1mg/kg in Figure 4A. These data also includes values that are non-normally distributed and close to outlier-like values. Was this finding replicated in a second cohort?

It is indeed the case that the only dopamine receptor agonist dose tested that significantly increased dSPN ensemble size was the 1 mg/kg data set, and that two data points disproportionately contributed to its mean. Importantly, the data presented in **Figure 4A** consist of two independent cohorts of mice imaged 4 months apart and the extreme values belong to separate mice from each cohort. Even after excluding both extreme values from the dataset, treatment with D1/2R agonists still significantly increased dSPN ensemble size compared to baseline (p = 0.0156, n = 7 FOVs; Wilcoxon signed rank test).

However, we wish to correct the Reviewer on an important point: the 1 mg/kg agonist data point is not the only evidence for an inverted U-shaped distribution. First, we note that the fact that blocking and stimulating D1/2 receptors both reduce dSPN ensemble size is, in and of itself, supportive of an inverted U distribution, even in the absence of an overt increase in dSPN ensemble size with low dose agonists. Second, we provide evidence that elevating endogenous dopamine levels with natural rewards and dopamine transporter blockers respectively increases and decreases the size of dSPN ensembles (Figure 4E-G), further supporting an inverted U-shaped relationship between dSPN ensemble size and dopamine signaling.

2. The authors focused on the analysis of ensemble size, firing rate and frequency of activated SPNs during forward locomotion. Presumably, locomotion arises from collaborative activation of certain SPN subsets and suppression of other subsets. Data in rearward locomotion and transitions should be readily available. Its evaluation would be important to bolster the findings, or reveal unexpected variances to the current interpretations.

Mice were habituated to head-fixation in the recording apparatus on both freely-rotating and motorized circular treadmills prior to experimentation. Once habituated, mice engaged in bouts of forward locomotion spontaneously on the freely-rotating treadmill and rarely (if ever) initiated rearward treadmill rotations. In fact, we interpreted rearward locomotion as an overt sign of behavioral distress, and thus prolonged habituation (and refrained from initiating imaging experiments) in mice that displayed such behavior. In addition, we never imposed rearward locomotion on the motorized treadmill. For these reasons, we do not have rearward locomotion data available for analysis.

We do, however, have data during transitions from immobility to forward locomotion and back to immobility. We report on these data in Figures 2B,C for Ca^2+^ event frequency and 5H for the percentage of imaged SPN active per 200 ms time bin. We also include in Author response image 1 an additional display of the percentage of imaged SPNs displaying Ca^2+^ transients as mice transition between immobility and forward locomotion in the dopamine-intact and dopamine-depleted (acutely and chronically) striatum:

**Author response image 1. sa2fig1:** 

These data are consistent with the changes in total SPN ensemble size reported in our manuscript. However, because measures of the instantaneous percentage of active SPNs cannot unambiguously distinguish between changes in overall active ensemble size (measured over an entire imaging session) and Ca^2+^ event frequency per active SPN, we refrained from displaying them in the manuscript for clarity.

3. Similarly, when only "total locomotion" is presented as the outcome measure, sub analyses should be performed on richer features of the movement data and presented/discussed to ensure that differential findings are not equally or better explained by differences in forward, reverse, or stalled locomotion or velocity.

We agree with the Reviewer that studying how neural activity in motor-related brain regions contributes to movement benefits from drawing comparisons with many behavioral parameters. But our study does not aim to explain what individual SPNs or groups of SPNs encode, or how they affect behavior. Instead, we sought to reveal how dopamine modulates striatal activity, while keeping as many experimental variables constant, including behavior. For this reason, we opted for a single, low-dimensional behavior (forward locomotion), which prior work shows is one of many ‘behavioral syllables’ spontaneously displayed by mice that is associated with the activation of a specific group of SPNs (Klaus et al., 2017; Markowitz et al., 2018).

We do, however, firmly believe that the reported changes in ensemble size do not stem from differences in velocity or forward, reverse, or stalled locomotion. First, as mentioned above, mice in our assay do not engage in rearward locomotion. Second, our manipulations did not shorten locomotor bout duration, indicating that stalled locomotion is unlikely to contribute to the observed effects on SPN ensemble size. Third, behavioral variables were not consistently affected by pharmacological manipulations (see Figure 3—figure supplement 1A–C, Figure 4—figure supplement 1A–C and Figure 5—figure supplement 1E–G). Fourth, we controlled for differences in locomotion frequency, duration and velocity by performing (Figures 3, 4 and 6) or repeating (Figure 5—figure supplement 2A) manipulations on motorized treadmills imposing a fixed number of locomotor bouts at set velocities. Fifth, dopamine-mediated changes in the percentage of imaged SPNs that are active during spontaneous locomotion were observed at all recorded velocities (Figure 5H). Lastly, the fact that diverse manipulations (ranging from unilateral lesion of nigrostriatal afferents to systemic DA receptor antagonists/agonists and natural rewards) produced consistent, *differential* effects on dSPN and iSPN ensembles despite variable effects on behavior strongly suggests that the cellular phenotypes we uncover here reflect the actions of dopamine, as opposed to being indirectly inherited from behavior.

Reviewer #2 (Recommendations for the authors):1. While D1 receptors, at least within the striatum, are relatively selective in their expression on dSPNs, D2 antagonists or agonists may act on iSPNs themselves, or local microcircuit elements, including cholinergic interneurons, dopamine terminals, and potentially local glutamatergic and GABAergic synapses. The authors allude to the glutamatergic inputs in the discussion, but one question in my mind was how much of the results here could be related to D2 autoreceptors on dopamine terminals. For example, D2 antagonists presynaptically increase striatal dopamine release, and D2 agonists decrease dopamine release (Palij et al., 1990 and others). At the very least, it would be worth discussing the potential mechanisms of the pharmacology.

The Reviewer is correct that dopamine receptors localize to many cells and processes in striatum in addition to dSPNs and iSPNs, making it difficult to attribute the effects of dopamine receptor agonists and antagonists to any one element. We first wish to clarify that the intent of this study is to reveal the net effect of dopamine modulation on striatal output in vivo, not to uncover the precise circuit elements or specific receptor types through which it does so. For this reason, we broadly stimulated or blocked all dopamine receptors simultaneously using cocktails of high-affinity D1/2R agonists or antagonists, respectively. One advantage of this approach is that we can a priori exclude confounds arising from secondary changes in presynaptic release of dopamine, since dopamine receptors are already pharmacologically blocked or stimulated. For instance, a presynaptic increase in dopamine release following D2R blockade ought to be of little consequence if all dopamine receptors in striatum are already blocked. Conversely, a decrease in dopamine release following D2R activation may be less impactful on SPNs whose receptors are coincidentally stimulated with D1/2 agonists.

We nevertheless performed experiments to mitigate concerns that modulation of presynaptic D2Rs significantly contributes to the observed effects. First, we replicated the contraction of dSPN ensembles observed with high dose D1/2R agonists (**Figure 4A**) in a small cohort of 3 mice treated with the D1R agonist SKF81297 alone at 3 mg/kg:

Second, the differential effects of D1/2R antagonists on the size of dSPN and iSPN ensembles (e.g. contraction and expansion, respectively) were undistinguishable from those observed 24 h following the loss of SNc neurons with 6-OHDA, suggesting that presynaptic modulation of DA release via D2 autoreceptors does not significantly contribute to the observed changes in SPN ensemble size.

2. Related to the above comment, though some of the differences between healthy and 6-OHDA treated animals in their response to dopamine drugs may be related to chronic changes in synaptic connectivity or intrinsic properties, some may relate to the loss of one of the targets of these drugs: the dopamine terminals. Might the effects of D2 antagonists in healthy animals be an interplay between direct effects on SPNs themselves, plus changes in dopamine release mediated by actions on dopamine terminals? And one of the reasons it's so different in 6-OHDA treated animals is that now this element has been removed?

As discussed above, despite the strong influence of D2 autoreceptors on dopamine release from SNc axons, we do not have evidence that this modulation contributes to the effects of D1/2R antagonists or 6-OHDA lesions on the size of dSPN ensembles.

3. The authors refer to L-DOPA-induced dyskinesia in their discussion, and compare their findings to published work with L-DOPA-induced dyskinesia, but this behavioral phenomenon was not mentioned/quantified in the text. Based on my understanding of the literature, the severity of dopamine depletion they induced (Figure 5B/C) combined with a dose of 10 mg/kg of L-DOPA would likely induce dyskinesia (eg Smith et al., 2012). Did their animals show dyskinesia? And if so, how did this relate to the patterns of neural activity reported?

We unfortunately did not perform detailed behavioral analyses using the validated Abnormal Involuntary Movement (AIM) scale. Axial, forelimb and orolingual AIMs typically develop in mice with unilateral dopamine neuron lesions upon chronic treatment with levodopa (at 3-25 mg/kg; Pavon et al., 2006; Smith et al., 2012; Cenci, 2014; Sebastianutto et al., 2016). We selected an intermediate dose and treated our mice only once, as we wished to restore striatal dopamine while avoiding severe dyskinetic episodes that may prevent mice from locomoting on the treadmill. We imaged striatal activity prior to levodopa, then administered levodopa before monitoring turning bias in a closed-arm plus-maze to confirm levodopa’s effectiveness, and finally imaged striatal activity again. The strong contraversive turning bias observed (see Figure 6—figure supplement 1A) is reminiscent of axial AIMs. However, we did not notice forelimb or orolingual AIMs, suggesting that our mice are unlikely to qualify as dyskinetic according to established standards. In agreement with this, our mice were able to locomote on the treadmill while head-fixed without apparent difficulty. That said, we agree with the Reviewer that it would be interesting for future investigations to relate SPN ensemble size to AIM scores to determine whether the dramatic expansion in dSPN ensemble size predisposes, or directly contributes to dyskinesia.

4. For interpretability and clarity, it would be helpful to include details of some major methods in the Results text and/or main Figure legends. For example, the authors should indicate in the Results and/or Figure legends that they used both an A2a-Cre x Ai14 cross and a viral approach to directly label iSPNs, that they used a cocktail of SCH23390 and raclopride to block D1/2 type dopamine receptors, or a cocktail of SKF81297 and quinpirole to activate D1/2 type dopamine receptors.

As suggested, the revised version of the manuscript specifies in the Results section that we used transgenic (A2a-Cre::Ai14 mice) as well as viral approaches (A2A-Cre with Cre-dependent tdTomato AAV) to label iSPNs in striatum (lines 125-129). The specific D1 and D2 receptor agonists and antagonists administered are now also indicated in the Results session (lines 191 and 222 in the revised manuscript).

5. It is unclear between the Figure and methods text what concentration of L-DOPA was used – the figure itself has "10" listed underneath (Figure 6E-G), but the methods list 12 mg/kg (which may instead refer to benserazide dosing?).

We apologize for this typo. The figure reflected the correct L-DOPA dose (10 mg/kg), while the Methods section erroneously listed 12 mg/kg, which the Reviewer rightly guessed referred to benserazide dosage. This error has now been corrected.